# Control Graph as Unified IO
# for Morphology-Task Generalization

**Hiroki Furuta**[1,2*]  **Yusuke Iwasawa**[1]  **Yutaka Matsuo**[1]  **Shixiang Shane Gu**[2,1]
[1]The University of Tokyo    [2]Google Research, Brain Team
furuta@weblab.t.u-tokyo.ac.jp

## Abstract

The rise of generalist large-scale models in natural language and vision has made us expect that a massive data-driven approach could achieve broader generalization in other domains such as continuous control. In this work, we explore a method for learning a single policy that manipulates various forms of agents to solve various tasks by distilling a large amount of proficient behavioral data. In order to align input-output (IO) interface among multiple tasks and diverse agent morphologies while preserving essential 3D geometric relations, we introduce *control graph*, which treats observations, actions and goals/task in a unified graph representation. We also develop MxT-Bench for fast large-scale behavior generation, which supports procedural generation of diverse morphology-task combinations with a minimal blueprint and hardware-accelerated simulator. Through efficient representation and architecture selection on MxT-Bench, we find out that a control graph representation coupled with Transformer architecture improves the multi-task performances compared to other baselines including recent discrete tokenization, and provides better prior knowledge for zero-shot transfer or sample efficiency in downstream multi-task imitation learning. Our work suggests large diverse offline datasets, unified IO representation, and policy representation and architecture selection through supervised learning form a promising approach for studying and advancing morphology-task generalization[1].

## 1 Introduction

The impressive success of large language models [25, 84, 8, 10, 19] has encouraged the other domains, such as computer vision [85, 42, 3, 52] or robotics [1, 51], to leverage the large-scale pre-trained model trained with massive data with unified input-output interface. These large-scale pre-trained models are innately multi-task learners: they surprisingly work well not only in the fine-tuning or few-shot transfer but also in the zero-shot transfer settings [86, 15]. Learning a "generalist" model seems to be an essential goal in the recent machine learning paradigm with the same key ingredients: curate **massive diverse dataset**, define **unified IO representation**, and perform **efficient representation and architecture selection**, altogether for best generalization.

In reinforcement learning (RL) for continuous control, various aspects are important for generalization. First, we care about "task" generalization. For instance, in robotic manipulation, we care the policy to generalize for different objects and target goal positions [58, 4, 103, 72]. Recent advances in vision and language models also enable task generalization through compositional natural language instructions [54, 91, 1, 23]. However, to scale the data, equally important is "morphology" generalization, where a single policy can control agents of different embodiment [100, 76] and can thereby ingest experiences from as many robots in different simulators [31, 96, 22] as possible. Most

---

[*]Work done as Student Researcher at Google.
[1]https://sites.google.com/view/control-graph

Offline Reinforcement Learning Workshop at Neural Information Processing Systems, 2022.

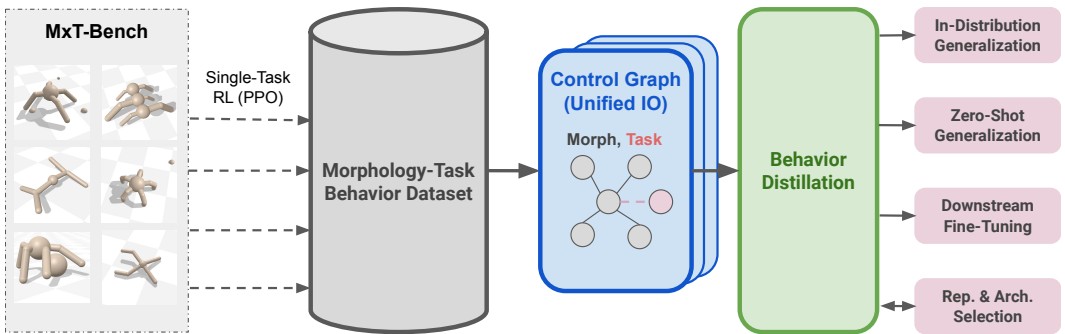

Figure 1: Behavior distillation pipeline. We first train a single-task policy for each environment on MxT-Bench, and then collect proficient morphology-task behavior dataset. To enable a single policy to learn multiple tasks and morphologies simultaneously, we convert stored transitions to the control graph representation to align with unified IO interface for multi-task distillation. After behavior distillation, the learned policy can be utilized for in-distribution or zero-shot generalization, downstream fine-tuning, and representation and architecture selection.

prior works [75, 44] only address either the task or morphology axis separately, and achieving broad generalization over task and morphology jointly remains a long-standing problem.

This paper first proposes *MxT-Bench*[2], the first multi-morphology and multi-task benchmarking environments, as a step toward building the **massive diverse dataset** for continuous control. MxT-Bench provides various combinations of different morphologies (ant, centipede, claw, worm, and unimal [44]) and different tasks (reach, touch, and twisters). MxT-Bench is easily scalable to additional morphologies and tasks, and is built on top of Brax [31] for fast behavior generation.

Next, we define **unified IO representation** for an architecture to ingest all the multi-morphology multi-task data. Inspired by *scene graph* [55] in computer vision that represents the 3D relational information of a scene, and by *morphology graph* [100, 13, 49, 44] that expresses an agent's geometry and actions, we introduce the notion of *control graph (CG)* as a unified interface to encode observations, actions, and goals (i.e. tasks) as nodes in the shared graph representation. Goals are represented as sub-nodes, and different tasks correspond to different choices: touching is controlling a torso node, while reaching is controlling an end-effector node (Figure 3). In contrast to discretizing and tokenizing every dimension as proposed in recent work [53, 87], this unified IO limits data representation it can ingest, but strongly preserves 3D geometric relationships that are crucial for any physics control problem [100, 38], and we empirically shows it outperforms naive tokenization in our control-focused dataset.

Lastly, while conventional multi-task or meta RL studies generalization through on-policy joint training [103, 20], we perform **efficient representation and architecture selection**, over 11 combinations of unified IO representation and network architectures, and 8 local node observations, for optimal generalization through *behavior distillation* (Figure 1), where RL is essentially treated as a (single-task, low-dimensional) behavior generator [41] and multi-task supervised learning (or offline RL [33]) is used for imitating all the behaviors [93, 14, 87]. Through offline distillation, we controllably and tractably evaluate two variants of CG representation, along with multiple network architectures (MLP, GNN [60], Transformers [98]), and show that CGv2 variant with Transformer improves the multi-task goal-reaching performances compared to other possible choices by 23% and provides better prior knowledge for zero-shot generalization (by 14∼18%) and fine-tuning for downstream multi-task imitation learning (by 50 ∼ 55 %).

As the fields of vision and language move toward broad generalization [18, 8], we hope our work could encourage RL and continuous control communities to continue growing diverse behavior datasets, designing different IO representations, and iterating more representation and architecture selection, and eventually optimize a single policy that can be deployed on any morphology for any task. In summary, our key contributions are:

---

[2]Pronounced as "mixed"-bench. It stands for "Morphology × Task".

- We develop *MxT-Bench*[3] as a test bed for *morphology-task generalization* with fast expert behavior generator. MxT-Bench supports the scalable procedural generation of both agents and tasks with minimal blueprints.

- We introduce *control graph*, a universal IO for control which treats the agent's observations, actions and goals/tasks in a unified graph representation, while preserving the task structure.

- We study generalization through offline supervised behavior distillation, where we can efficiently try out various design choices; over 11 combinations of unified IO representation and network architectures, and 8 local node observations. As a result, we find that Transformer with CGv2 achieves the best multi-task performances among other possible designs (MLP, GNN and Transformer with CGv1, Tokenized-CGv2, etc.) in both in-distribution and downstream tasks, such as zero-shot transfer and fine-tuning for multi-task imitation learning.

## 2  Related Work

**Morphology Generalization**  While, in RL for continuous control, the policy typically learns to control only a single morphology [94, 96], several works succeed in generalizing the control problem for morphologically different agents to solve a locomotion task by using morphology-aware Graph Neural Network (GNN) policies [100, 49]. In addition, several work [62, 44, 48, 97] have investigated the use of Transformer [98]. Other work jointly optimize the morphology-agnostic policy and morphology itself [80, 43, 105, 47], or transfer a controller over different morphologies [24, 13, 46, 71].

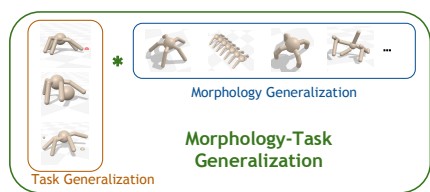

Figure 2: We tackle *morphology-task generalization*, which requires achieving both task and morphology generalization simultaneously.

While substantial efforts have been investigated to realize morphology generalization, those works mainly focus on only a single task (e.g. running), and less attention is paid to multi-task settings, where the agents attempt to control different states of their bodies to desired goals. We believe that goal-directed control is a key problem for an embodied single controller. Concurrently, Feng et al. [30] propose a RL-based single controller that is applied to different quadruped robots and target poses in the sim-to-real setting. In contrast, our work introduces the notion of control graph as a unified IO that represents observations, actions, and goals in a shared graph, and can handle more diverse morphologies to solve multiple tasks.

**Task Generalization**  In the previous research, task generalization has been explored in multi-task or meta RL literature [99, 27, 11, 95, 21, 67, 102, 63]. Each task might be defined by the difference in goals, reward functions, and dynamics [37, 59, 28], under shared state and action spaces. Some works leverage graph representation to embed the compositionality of manipulation tasks [68, 108, 69, 61, 38], while others use natural language representation to specify diverse tasks [54, 91, 1, 50, 23]. Despite the notable success of acquiring various task generalization, multi-task RL often deals with only a single morphology. We aim to extend the general behavior policy into the "cartesian product" of tasks and morphologies (as shown in Figure 2) to realize a more scalable and capable controller.

**Transformer for RL**  Recently, Chen et al. [12] and Janner et al. [53] consider offline RL as supervised sequential modeling problem and following works achieve impressive success [87, 64, 36, 101, 90, 107, 79]. In contrast, our work leverages Transformer to handle topological and geometric information of the scene, rather than a sequential nature of the agent trajectory.

**Behavior Distillation**  Due to the massive try-and-error and large variance, training a policy from scratch in online RL is an inefficient process, especially in multi-task setting. It is more efficient to use RL for generating single-task behaviors (often from low dimensions) [41] and then use supervised learning to imitate all behaviors with a large single policy [65, 93, 2, 14]. Several works tackle the large-scale behavior distillation with Transformer architecture [87, 64], or with representation that treats observations and actions in the same vision-language space [106, 92]. Our work utilizes

---

[3]https://github.com/frt03/mxt_bench

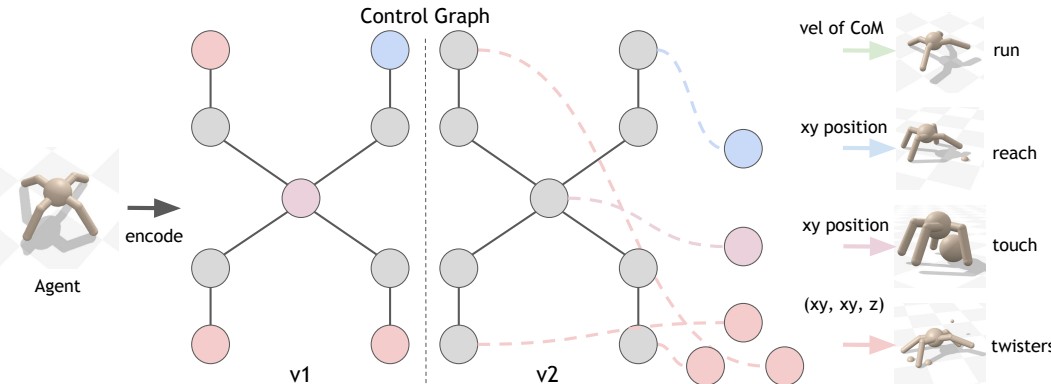

Figure 3: We propose the notion of *control graph*, which expresses the agent's observations, actions, and goals/tasks in an unified graph representation, while preserving the geometric structure of the task. We develop two practical implementations; **control graph v1** (left) accepts the morphological graph, encoded from the agent's geometric information, as an input-output interface, and merges positional goal information as a part of corresponding node features. **control graph v2** (right) treats given goals as extra disjoint nodes of morphological graph. While most of prior morphology-agnostic RL have focused on locomotion (run task), i.e. a single static goal node controlling (maximizing) the velocity of center of mass, control graph could naturally extend morphology-agnostic control to other goal-oriented tasks: single static goal node for reach, multiple static goal nodes for twisters, and single dynamic goal node tracking a movable ball for an object interaction task; touch.

similar pipeline, but focuses on finding the good representation and architecture to generalize across morphology and tasks simultaneously with proposed control graph.

## 3 Preliminaries

In RL, consider a Markov Decision Process with following tuple $(\mathcal{S}, \mathcal{A}, p, p_1, r, \gamma)$, which consists of state space $\mathcal{S}$, action space $\mathcal{A}$, state transition probability function $p : \mathcal{S} \times \mathcal{A} \times \mathcal{S} \to [0, \infty)$, initial state distribution $p_1 : \mathcal{S} \to [0, \infty)$, reward function $r : \mathcal{S} \times \mathcal{A} \to \mathbb{R}$, and discount factor $\gamma \in [0, 1)$. Deep RL parameterizes a Markovian policy $\pi : \mathcal{S} \to \mathcal{A}$ with neural networks.The agent seeks optimal policy $\pi^*$ that maximizes the discounted cumulative rewards:

$$\pi^* = \arg\max_\pi \frac{1}{1 - \gamma} \mathbb{E}_{s \sim \rho^\pi(s), a \sim \pi(\cdot|s)} \left[ r(s, a) \right], \tag{1}$$

where $p_t^\pi(s_t) = \iint_{s_{0:t}, a_{0:t-1}} \prod_t p(s_t|s_{t-1}, a_{t-1})\pi(a_t|s_t)$ and $\rho^\pi(s) = (1 - \gamma) \sum_t \gamma^t p_t^\pi(s_t = s)$ represent time-aligned and time-aggregated state marginal distributions following policy $\pi$.

### 3.1 Graph Representation for Morphology-Agnostic Control

Following prior continuous control literature [96], we assume the agents have bodies modeled as simplified skeletons of animals. An agent's morphology is characterized by the parameter for rigid body module (torso, limbs), such as radius, length, mass, and inertia, and by the connection among those modules (joints). In order to handle such geometric and topological information, an agent's morphology can be expressed as an acyclic tree graph representation $\mathbf{G} := (\mathbf{V}, \mathbf{E})$, where $\mathbf{V}$ is a set of nodes $v^i \in \mathbf{V}$ and $\mathbf{E}$ is a set of edges $e^{ij} \in \mathbf{E}$ between $v^i$ and $v^j$. The node $v^i$ corresponds to $i$-th module of the agent, and the edge $e^{ij}$ corresponds to the hinge joint between the nodes $v^i$ and $v^j$. Each joint may have 1-3 actuators corresponding to a degree of freedom. If a joint has several actuators, the graph $\mathcal{G}$ is considered a multipath graph. This graph-based formulation can describe the various agents' morphologies in a tractable manner [100, 49].

We assume that node $v^i$ observes local sensory input $s_t^i$ at time step $t$, which includes the information of limb $i$ such as position, velocity, orientation, joint angle, or morphological parameters. To process these node features and graph structure, a morphology-agnostic policy can be modeled as node-based GNN [60, 6], which takes a set of local observations $\{s_t^i\}_{i=1}^{|\mathbf{V}|}$ as an input and emit the actions for actuators of each joint $\{a_t^e\}_{e=1}^{|\mathbf{E}|}$. The objective function of morphology-agnostic RL is the average of Equation 1 among given morphologies.

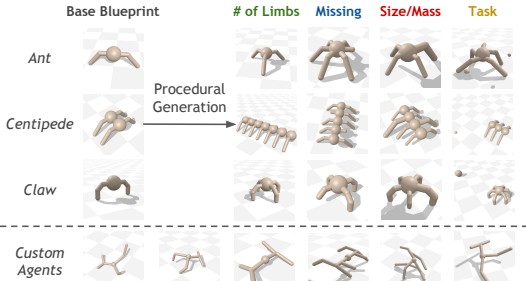

| Benchmark | Multi-Task | Multi-Morphology | Scalability |
|---|---|---|---|
| MuJoCo & DM Control | ✗ | ✗ | ✗ |
| Meta-World [103] | ✓ | ✗ | ✗ |
| Huang et al. [49] | ✗ | ✓ | ✗ |
| Gupta et al. [44] | ✗ | ✓ | ✓ |
| **MxT-Bench(Ours)** | ✓ | ✓ | ✓ |

Figure 4: The overview of MxT-Bench, which can procedurally generate both various morphologies and tasks with minimal blueprints. MxT-Bench can not only construct the agents with different number of limbs, but also randomize missing limbs and size/mass of bodies. We could design the tasks with parameterized goal distributions. It also supports to import custom complex agents such as unimals [44]. Compared to relevant RL benchmarks in terms of (1) multi-task (task coverage), (2) multi-morphology (morphology coverage), and (3) scalability, MuJoCo [96] and DM Control [94] only have a single morphology for a single task. While other existing works [103, 49] partially cover task-/morphology-axis with some sort of scalability, they do not satisfy all criteria.

## 3.2 Goal-conditional RL

In goal-conditional RL [57, 88], the agent aims to find an optimal policy $\pi^*(a|s, s_g)$ conditioned on goal $s_g \in \mathcal{G}$, where $\mathcal{G}$ stands for goal space that is sub-dimension of state space $\mathcal{S}$ (e.g. XYZ coordinates, velocity, or quaternion). The desired goal $s_g$ is sampled from the given goal distribution $p_\psi : \mathcal{G} \to [0, \infty)$, where $\psi$ stands for task in the task space $\Psi$ (e.g. reaching the agent's leg or touching the torso to the ball). The reward function can include a goal-reaching term, that is often modeled as $r_\psi(s_t, s_g) = -d_\psi(s_t, s_g)$, where $d_\psi(\cdot, \cdot)$ is a task-dependent distance function, such as Euclidean distance, between the sub-dimension of interest in current state $s_t$ and given goal $s_g$. Some task $\psi$ give multiple goals to the agents. In that case, we overload $s_g$ to represent a set of goals; $\{s_g^i\}_{i=1}^{N_\psi}$, where $N_\psi$ is the number of goals that should be satisfied in the task $\psi$.

## 3.3 Morphology-Task Generalization

This paper aims to achieve morphology-task generalization, where the learned policy should generalize over tasks and morphologies simultaneously. The optimal policy should generalize over morphology space $\mathcal{M}$, task $\Psi$, and minimize the distance to any given goal $s_g \in \mathcal{G}$. Mathematically, this objective can be formulated as follows:

$$\pi^* = \arg\max_\pi \frac{1}{1-\gamma} \mathbb{E}_{m,\psi\sim\mathcal{M},\Psi} \left[ \mathbb{E}_{s_g\sim p_\psi(s_g)} \left[ \mathbb{E}_{\boldsymbol{s}^m,\boldsymbol{a}^m\sim\rho^\pi(\boldsymbol{s}^m),\pi(\cdot|\boldsymbol{s}^m,s_g)} \left[ -d_\psi(\boldsymbol{s}^m, s_g) \right] \right] \right], \quad (2)$$

where the graph representation of morphology $m \in \mathcal{M}$ is denoted as $\mathbf{G}_m = (\mathbf{V}_m, \mathbf{E}_m)$, and $\boldsymbol{s}^m := \{s_t^i\}_{i=1}^{|\mathbf{V}_m|}$ and $\boldsymbol{a}^m := \{a_t^e\}_{e=1}^{|\mathbf{E}_m|}$ stand for a set of local observations and actions of morphology $m$. While we can use multi-task online RL to maximize Equation 2 in principle, it is often sample inefficient due to the complexity of task, which requires a policy that can handle the diversity of the scene among morphology $\mathcal{M}$, task $\Psi$, and goal space $\mathcal{G}$ simultaneously.

## 4 Method

### 4.1 MxT-Bench as a Test Bed for Morphology-Task Generalization

To overcome these shortcomings in the existing RL environments, we develop *MxT-Bench*, which has a wide coverage over both tasks and morphologies to test morphology-task generalization, with the functionalities for procedural generation from minimal blueprints (Figure 4). MxT-Bench is built on top of Brax [31] and Composer [41], for faster iteration of behavior distillation with hardware-accelerated environments. Beyond supporting multi-morphology and multi-task settings, the scalability of MxT-Bench helps to test the broader range of morphology-task generalization since we can easily generate out-of-distribution tasks and morphologies, compared to manually-designed morphology or task specifications.

In the morphology axis, we prepare 4 types of blueprints (ant, claw, centipede, and worm) as base morphologies, since they are good at the movement on the XY-plane. Through MxT-Bench, we can easily spawn agents that have different numbers of bodies, legs, or different sizes, lengths, and weights. Moreover, we can also import the existing complex morphology used in previous work. For instance, we include 60+ morphologies that are suitable for goal-reaching, adapted from Gupta et al. [44] designed in MuJoCo. In the task axis, we design reach, touch, and twisters [4] as basic tasks, which could evaluate different aspects of the agents; the simplest one is the reach task, where the agents aim to put their leg on the XY goal position. In the touch task, agents aim to create and maintain contact between a specified torso and a movable ball. The touch task requires reaching behavior, while maintaining a conservative momentum to avoid kicking the ball away from the agent. Twisters tasks are the multi-goal problems; for instance, the agents should satisfy XY-position for one leg, and Z height for another leg. We pre-define 4 variants of twisters with max 3 goals (see Appendix C.2 for the details). Furthermore, we could easily specify both initial and goal position distribution with parameterized distribution. In total, we prepare 180+ environments combining the morphology and task axis for the experiments in the later section. See Appendix C for further details.

## 4.2  Behavior Distillation

Toward broader generalization over morphologies and tasks, a single policy should handle the diversity of the scene among morphology $\mathcal{M}$, task $\Psi$, and goal space $\mathcal{G}$ simultaneously. Multi-task online RL from scratch, however, is difficult to tune, slow to iterate, and hard to reproduce. Instead, we employ behavior cloning on RL-generated expert behaviors to study morphology-task generalization. To obtain rich goal-reaching behaviors, we train a single-morphology single-task policy using PPO [89] with a simple MLP policy, which is significantly more efficient than multi-morphology training done in prior work [44, 62, 49]. Since MxT-Bench is built on top of Brax [31], a hardware-accelerated simulator, training PPO policies can be completed in about 5~30 minutes per environment (on NVIDIA RTX A6000). We collect many behaviors per morphology-task combination from the expert policy rollout. We then train a single policy $\pi_\theta$ with a supervised learning objective:

$$\mathcal{L}_\pi = -\mathbb{E}_{m,\psi \sim \mathcal{M}, \Psi} \left[ \mathbb{E}_{\boldsymbol{s}^m, \boldsymbol{a}^m, s_g \sim \mathcal{D}_{m,\psi}} \left[ \log \pi_\theta(\boldsymbol{a}^m | \{\boldsymbol{s}^m, s_g\}) \right] \right],$$

where $\mathcal{D}_{m,\psi}$ is an expert dataset of morphology $m$ and task $\psi$. Importantly, offline behavior distillation protocol runs (parallelizable) single-task online RL only once, and allows us to reuse the same fixed data to try out various design choices, such as model architectures or local features of control graph, which is often intractable in multi-task online RL.

## 4.3  Control Graph

To learn a single policy that could solve various morphology-task problems, it is essential to unify the input-output interface among those. Inspired by the concept of scene graph [55] and morphological graph [100], we introduce the notion of control graph (CG) representation that incorporates goal information while preserving the geometric structure of the task. Control graph could express the agent's observations, actions, and goals/tasks in an unified graph space. Although most prior morphology-agnostic RL has focused on locomotion (running) with the reward calculated from the velocity of the center of mass, control graph formulation could naturally extend morphology-agnostic RL to multi-task goal-oriented settings: including static single positional goals (reaching), multiple-goal problems (twister-game) and object interaction tasks (ball-touching). In practice, we develop two different way to inform the goals and tasks (Figure 3); control graph v1 (CGv1) accepts the morphological graph, encoded from the agent's geometric information, as an input-output interface, and merges positional goal information as a part of corresponding node features. For instance, in touch task, CGv1 includes XY position of the movable ball as an extra node features of the body node. Moreover, control graph v2 (CGv2) considers given goals as additional disjoint nodes of morphological graph representation. These control graph strategies enable the policy to handle a lot of combination of tasks and morphologies simultaneously.

**Transformer for Control Graph**  While control graph could represent the agent and goals in a unified manner, because the task structure may change over time, the policy should unravel the implicit relationship between the agent's modules and goals dynamically. We mainly employ Transformer as a policy architecture because it can process morphological graph as a fully-connected graph

---

[4] https://en.wikipedia.org/wiki/Twister_(game)

| | Random | MLP | GNN (CGv1) | Transformer (CGv1) | Transformer (CGv2) | Token-CGv2 |
|---|---|---|---|---|---|---|
| **In-Distribution** | $1.2019 \pm 0.41$ | $0.5150 \pm 0.01$ | $0.4776 \pm 0.01$ | $0.4069 \pm 0.02$ | $\mathbf{0.3128 \pm 0.02}$ | $0.3402 \pm 0.01$ |
| **In-Distribution (unimal)** | $0.9090 \pm 0.03$ | $0.6703 \pm 0.01$ | – | $0.4839 \pm 0.02$ | $\mathbf{0.4178 \pm 0.01}$ | – |
| **Compositional (Morphology)** | $1.1419 \pm 0.41$ | $0.7216 \pm 0.01$ | – | $0.4940 \pm 0.01$ | $\mathbf{0.4066 \pm 0.01}$ | – |
| **Compositional (Task)** | $0.8932 \pm 0.01$ | $0.6849 \pm 0.01$ | – | $0.5395 \pm 0.04$ | $\mathbf{0.4461 \pm 0.05}$ | – |
| **Out-of-Distribution** | $0.8979 \pm 0.01$ | $0.7821 \pm 0.02$ | – | $0.6144 \pm 0.04$ | $\mathbf{0.5266 \pm 0.04}$ | – |

Table 1: The average normalized final distance in various types of morphology-task generalization on MxT-Bench. We test (1) in-distribution, (2) compositional morphology/task, and (3) out-of-distribution generalization. (2) and (3) evaluate zero-shot transfer. We compare MLP, GNN and Transformer with CGv1, Transformer with CGv2, and tokenized CGv2 [87]. CGv2 improves multi-task performance to other choices by 23 % in the in-distribution evaluation, and achieves better zero-shot transfer in the compositional and out-of-distribution settings by $14 \sim 18$ %.

and achieve notable performance by leveraging the hidden relationships between the node beyond manually-encoded geometric structure [62]. Transformer first encodes control graph to the latent representation vector $z_0$ with shared single-layer MLP and learnable position embedding (PE) (in case of CGv1, we omit $s_g$, but include it to corresponding node observation $s^i$ instead):

$$z_0 = [\mathrm{MLP}(s^1), \ldots, \mathrm{MLP}(s^{|V_m|}), \; \mathrm{MLP}(s_g)] + \; \mathrm{PE},$$

then multi-head attention (MHA) [98] and layer normalization (LayerNorm) [5] are recursively applied to latent representation $z_l$ at $l$-th layer,

$$z'_l = \mathrm{LayerNorm}(\mathrm{MHA}(z_{l-1}) + z_{l-1}),$$
$$z_l = \mathrm{LayerNorm}(\mathrm{MLP}(z'_l) + z'_l), \qquad (l = 1, ..., L).$$

Before decoding the action per module from the last-layer latent representation $z_L$, we employ the residual connection of the node features [62],

$$a^m = [\mathrm{MLP}([z_L^1, s^1]), \ldots, \; \mathrm{MLP}([z_L^{|V_m|}, s^{|V_m|}]), \; \mathrm{MLP}([z_L^{|V_m|+1}, s_g])],$$

where shared MLP has a single layer and tanh activation to clip the output within the range of [-1, 1]. We mask out the output from the goal modules or modules that have no actuators to ensure action size as $|E_m|$. See Appendix A for further details of implementation.

## 5 Experiments

We first evaluate the multi-task performance of control graph representation (CGv1, CGv2) in terms of in-distribution (known morphology and task with different initialization), compositional (known task with unseen morphology or known morphology with unseen task), and out-of-distribution generalization (either morphology or task is unseen) on MxT-Bench (Section 5.1). Then, we investigate whether control graph could contribute to obtaining better control prior for multi-task fine-tuning (Section 5.2). Lastly, we examine how control graph works well by visualizing attention weights (Section 5.3). The results are averaged among 4 random seeds. See Appendix A for the hyperparameters. We also examine the other axis of representations or architectures (Appendix B, F, G) and test the effect of dataset size, the number of morphology-task combinations, and model size (Appendix H).

**Evaluation Metric** Goal-reaching tasks are evaluated by the distance to the goals at the end of episode [82, 83, 39, 17, 29]. However, this can be problematic in our settings, because the initial distance or the degree of goal-reaching behaviors might be different among various morphologies and tasks. We measure the performance of the policy $\pi$ by using a normalized final distance metric $\bar{d}(\mathcal{M}, \Psi; \pi)$ over morphology $\mathcal{M}$ and task space $\Psi$ with pre-defined max/min value of each morphology $m$ and task $\psi$,

$$\bar{d}(\mathcal{M}, \Psi; \pi) := \frac{1}{|\mathcal{M}||\Psi|} \sum_m^{\mathcal{M}} \sum_\psi^{\Psi} \mathbb{E}_{s_g \sim p_\psi} \left[ \sum_{i=1}^{N_\psi} \frac{d_\psi(s_T^m, s_g^i) - d_{\min}^{i,m,\psi}}{d_{\max}^{i,m,\psi} - d_{\min}^{i,m,\psi}} \right], \qquad (3)$$

where $s_T^m$ is the last state of the episode, $d_{\max}^{i,m,\psi}$ is a maximum, and $d_{\min}^{i,m,\psi}$ is a minimum distance of $i$-th goal $s_g^i$ with morphology $m$ and task $\psi$. We use a distance threshold to train the expert PPO policy as $d_{\min}^{i,m,\psi}$, and average distance from initial position of the scene as $d_{\max}^{i,m,\psi}$. Equation 3 is normalized around the range of [0, 1] and the smaller, the better. See Appendix C for the details.

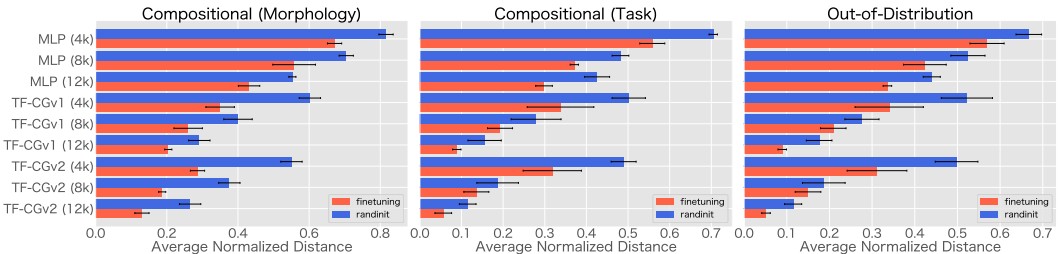

Figure 5: Multi-task goal-reaching performances on fine-tuning (multi-task imitation) for compositional and out-of-distribution evaluation. These results reveal that fine-tuning outperforms random initialization in all settings, and fine-tuned CGv2 outperforms others by $50 \sim 55$ %. See Appendix J for the detailed scores.

## 5.1 Behavior Distillation on MxT-Bench

We systematically evaluate three types of morphology-task generalization; in-distribution, compositional, and out-of-distribution generalization through behavior distillation. As baselines, we compare MLP, GNN [100] with CGv1, Transformer with CGv1 or CGv2, and tokenized CGv2, similar to Reed et al. [87] (see Appendix G for the details).

In in-distribution settings, we prepare 50 environments and 60 unimal environments adapted from Gupta et al. [44] (see Appendix C.3 for the details) for both training and evaluation. Compositional and out-of-distribution settings evaluate zero-shot transfer. In compositional settings, we test morphology and task generalization separately; we prepare 38 train environments and 12 test environments with hold-out morphologies for morphology evaluation, and leverage 50 train environments and prepare 9 test environments with unseen task for task evaluation. In out-of-distribution settings, we also leverage 50 environments as a training set, and define 27 environments with diversified morphologies and unseen task as an evaluation set. The proficient behavioral data contains 12k transitions per each environment. See Appendix E for the details of environment division.

Table 1 reveals that CGv2 achieves the best multi-task goal-reaching performances among other possible combinations in all the aspects of generalization. Comparing average normalized distance, CGv2 improves the multi-task performance against the second best, CGv1, by 23% in in-distribution evaluation. Following previous works [62, 44], Transformer with CGv1 achieves better goal-reaching behaviors than GNN. In compositional and out-of-distribution zero-shot evaluation, CGv2 outperforms other choices by $14 \sim 18\%$. Moreover, the compositional zero-shot performance of CGv2 is comparable with the performance of CGv1 in in-distribution settings. These results imply CGv2 might be the better formulation to realize the morphology-task generalization.

## 5.2 Does Control Graph Obtain Better Prior for Control?

To reveal whether the distilled policy obtains reusable inductive bias for unseen morphology or task, we test the fine-tuning performance for multi-task imitation learning on MxT-Bench. We adopt the same morphology-task division for compositional and out-of-distribution evaluation in Section 5.1. Figure 5 shows that fine-tuning outperforms random initialization in all settings, which suggests that behavior-distilled policy works as a better prior knowledge for control. The same as zero-shot transfer results in Section 5.1, CGv2 outperforms other baselines, and is better than the second best, CGv1 by $50 \sim 55\%$. Furthermore, CGv2 could work even with a small amount of dataset (4k, 8k); for instance, in compositional morphology evaluation (left in Figure 5), CGv2 trained with 8k transitions still outperforms competitive combinations with 12k transitions, which indicates better sample efficiency (see Appendix J for the detailed scores). These results suggest CGv2 significantly captures the structure of control graph as prior knowledge for downstream tasks.

## 5.3 How Does Control Graph Work Well?

The experimental results suggest, despite the slight difference, CGv2 generalizes various morphologies and tasks better than CGv1. To find out the difference between those, we qualitatively analyze the attention weights in Transformer. Figure 6 shows that CGv2 consistently focuses on goal nodes over time, and activates important nodes to solve the task; for instance, in ant_twisters, CGv2 firstly

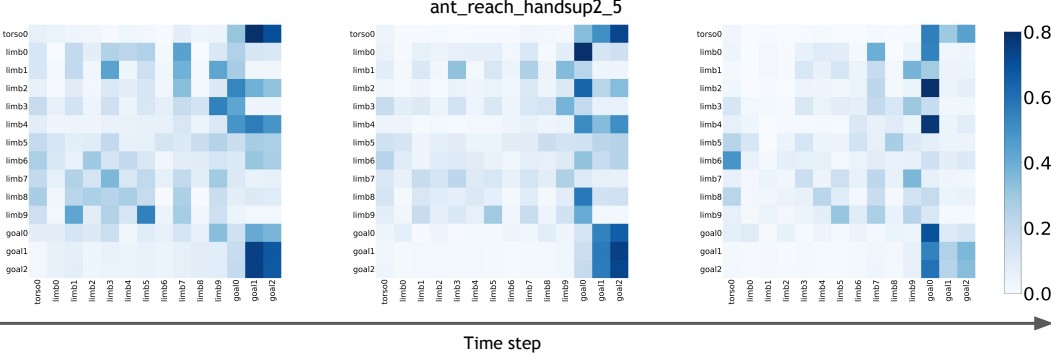

Figure 6: Attention analysis of CGv2 in ant_twisters. We visualize the attention weights of CGv2 during the rollout. CGv2 consistently focuses on goal nodes over time, and activates important nodes to solve the task.

tries to raise the agent's legs to satisfy goal1 and goal2, and then focus on reaching a leg to goal0. Temporally-consistent attention to goal nodes and dynamics attention to relevant nodes can contribute to generalization over goal-directed tasks and morphologies. In contrast, CGv1 does not show such consistent activation to goal-conditioned node, rather it demonstrates some periodic patterns as implied in prior works [62, 49]. See Appendix K for the full results.

## 6 Discussion and Limitation

While the experimental evaluation on MxT-Bench implies that control graph is a simple and effective method to distill the diverse proficient behavioral data into a generalizable single policy, there are some limitations. For instance, we focus on distillation from expert policies only, but it is still unclear whether control graph works with moderate or random quality behaviors in offline RL [33, 66, 32]. Combining distillation with iterative data collection [39, 74] or online fine-tuning [70] would be a promising future work. In addition, we avoided tasks where expert behaviors cannot be generated easily by single-task RL without fine-scale reward engineering or human demonstrations; incorporating such datasets or bootstrapping single-task RL from the distilled policy could be critical for scaling the pipeline to more complex tasks such as open-ended and dexterous manipulation [72, 38, 16]. Since control graph only uses readily accessible features in any simulator and could be automatically defined through URDFs or MuJoCo XMLs, in future work we aim to keep training our best control graph architecture policy on additional data from more functional and realistic control behaviors from other simulators like MuJoCo [103, 94], PyBullet [91], IsaacGym [16, 81], and Unity [56], and show it achieves better scaling laws than other representations [87] on broader morphology-task families.

## 7 Conclusion

The broader range of behavior generalization is a promising paradigm for RL. To achieve morphology-task generalization, we propose control graph, which expresses the agent's modular observations, actions, and goals as a unified graph representation while preserving the geometric task structure. As a test bed for morphology-task generalization, we also develop MxT-Bench, which enables the scalable procedural generation of agents and tasks with minimal blueprints. Fast-generated behavior datasets of MxT-Bench with RL allow efficient representation and architecture selection through supervised learning, and CGv2, variant of control graph, achieves the best multi-task performances among other possible designs (MLP, GNN and Transformer with CGv1, and tokenized-CGv2, etc), outperforming them in in-distribution evaluation (by 23 %), zero-shot transfer among compositional or out-of-distribution evaluation (by 14 ∼ 18 %) and fine-tuning for downstream multi-task imitation (50 ∼ 55 %). We hope our work will encourage the community to explore scalable yet incremental approaches for building a universal controller.

**Acknowledgments**

This work was supported by JSPS KAKENHI Grant Number JP22J21582. We thank Mitsuhiko Nakamoto and Daniel C. Freeman for converting several MuJoCo agents to Brax, So Kuroki for the support on implementations, Yujin Tang, Kamyar Ghasemipour, Yingtao Tian, and Bert Chan for helpful feedback on this work.

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

# Appendix

## A Details of Implementation

The hyperparameters we used are listed in Table 2. We implement MLP and Transformer policy with Jax [9] and Flax [45]. For the implementation of GNN policy, we use a graph neural network library, Jraph [40].

| Method | Hyperparameter | Value |
|---|---|---|
| Shared | Learning rate | 3e-4 |
| | Batch size | 256 |
| | Gradient clipping | 0.1 |
| | Activation function | ReLU |
| | Gradient steps | 100k |
| MLP | Hidden size | 1024 |
| | # of layers | 2 |
| Transformer | Embedding size | 256 |
| | Attention hidden size | 512 |
| | # of attention heads | 2 |
| | # of attention layers | 3 |
| GNN | Hidden size | 256 |
| | # of layers (per node) | 3 |

Table 2: Hyperparameters for each method.

## B Architecture Selection: Position Embedding

As a part of architecture selection, we investigate whether position embedding (PE) contributes to the generalization. Our empirical results in Table 3 suggest that, multi-task goal reaching performance seems comparable between those. However, in more diverse morphology domains, PE plays an important role. Therefore, we include PE into a default design.

| Sub-domain | Transformer (CGv2) (w/ PE) | Transformer (CGv2) (w/o PE) |
|---|---|---|
| ant_reach | $0.3206 \pm 0.06$ | $0.3966 \pm 0.08$ |
| ant_touch | $0.2668 \pm 0.08$ | $0.4573 \pm 0.14$ |
| ant_twisters | $0.1039 \pm 0.05$ | $0.1569 \pm 0.02$ |
| claw_reach | $0.3581 \pm 0.04$ | $0.3508 \pm 0.03$ |
| claw_touch | $0.2573 \pm 0.08$ | $0.3278 \pm 0.07$ |
| claw_twisters | $0.3442 \pm 0.04$ | $0.3071 \pm 0.04$ |
| centipede_reach | $0.1057 \pm 0.04$ | $0.0610 \pm 0.02$ |
| centipede_touch | $0.3869 \pm 0.04$ | $0.1687 \pm 0.03$ |
| worm_touch | $0.8952 \pm 0.05$ | $0.8427 \pm 0.03$ |
| **Average Dist.** | $0.3128 \pm 0.02$ | $0.3142 \pm 0.03$ |
| unimal_reach | $0.4532 \pm 0.01$ | $0.5856 \pm 0.02$ |
| unimal_touch | $0.4461 \pm 0.05$ | $0.3799 \pm 0.06$ |
| unimal_twisters | $0.3540 \pm 0.02$ | $0.3236 \pm 0.04$ |
| **Average Dist.** | $0.4178 \pm 0.01$ | $0.4297 \pm 0.03$ |

Table 3: The average normalized final distance on in-distribution evaluation. We compare the effect of position embedding.

# C   Details of MxT-Bench

## C.1   Morphology

We prepare 4 base blueprints; ant, centipede, claw, and worm, for procedural scene generation (Figure 7). Base ant has 1 torso and 2 legs with 2 joints per limb, and we could procedurally generate the agents with different number of limbs. Base centipede has 2 bodies and 4 legs with 2 joints per limb, and we could procedurally generate the agents with different number of bodies. Base claw has 1 torso and 2 legs with 4 joints per limb (each leg consists of 3 modules), and we could procedurally generate the agents with different number of limbs. Base worm has 2 bodies and no legs, and we could procedurally generate the agents with different number of bodies.

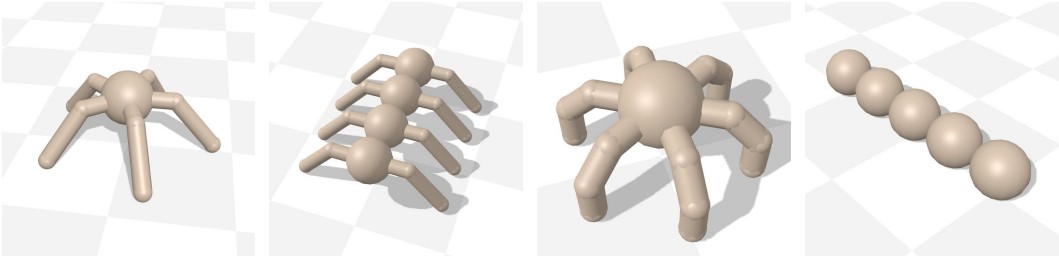

Figure 7: Examples of procedurally-generated morphology from base blueprints in MxT-Bench. From left to right, each figure shows the example of 5-leg ant, 4-body centipede, 6-leg claw, and 5-body worm.

Furthermore, we develop the functionality for morphology diversification, with missing, mass, and size parameters (Figure 8). Missing randomization lacks one module at one leg. This might be an equivalent situation that one leg is broken. Mass randomization changes the default mass of each module with specified scales. The appearance does not change, but certainly the dynamics would differ. Size randomization changes the default length and radius of each module with specified scales.

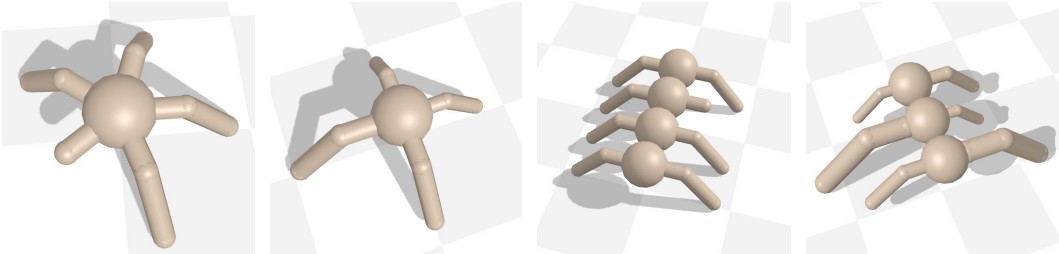

Figure 8: Examples of diversified morphology from base blueprints in MxT-Bench. From left to right, each figure shows the example of 5-leg-1-missing ant, 5-leg-size-randomized ant, 4-body-1-missing centipede, and 3-body-size-randomized centipede.

## C.2   Task

We prepare 4 base tasks with parameterized goal distributions; reach, touch, twisters, and push, for procedural task generation (Figure 9). Reach task requires the agents to put their one leg to the given goal position (XY). The variant, reach_hard task, represents that the goal distribution is farther than reach task. Touch task requires the agents to contact their body or torso to the movable ball (i.e. movable ball is an goal). Twisters is a multi-goal problem; the agents should satisfy given goals at the same time. There are two basic constraints; reach and handsup. Handsup requires the agents to raise their one leg to the given goal Z height. Twisters has some combinations like reach_handsup, reach_hard_handsup, reach2_handsup, or reach_handsup2. For instance, in reach_handsup2, the agents should put their one leg to the given goal position, and raise their two legs to the given goal heights simultaneously. Push task requires the agents to move the box object to the given goal position (XY). Since it has richer interaction with the object, push might be more difficult task than those three. We use this task in Appendix L.

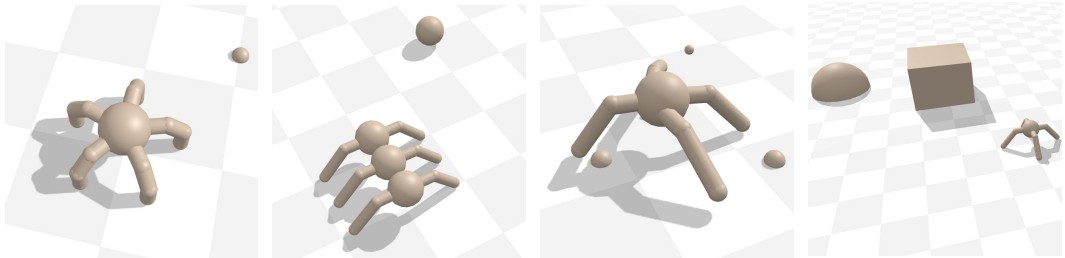

Figure 9: Examples of pre-defined task in MxT-Bench. From left to right, each figure shows the example of reach, touch, twisters, and push task.

## C.3 Custom Morphology

MxT-Bench also supports custom morphology import used in previous work. For instance, Gupta et al. [44] propose unimal agents that are generated via evolutional strategy and designed for MuJoCo. Since they are not manually designed, their morphologies seem more diverse than our ant, centipede, claw, and worm. We inspect unimals whether they are suitable for goal-reaching, and include 72 morphologies from there (and use 60 morphologies for the experiments). Figure 10 shows some example of unimals.

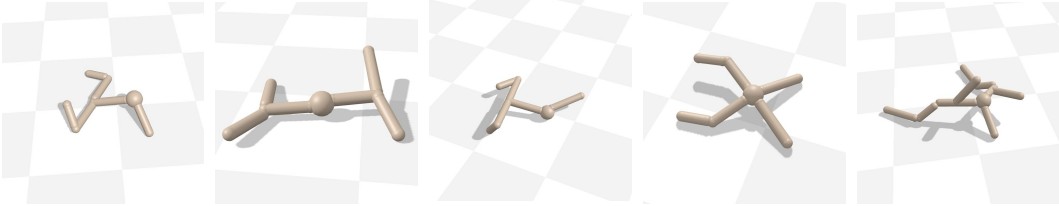

Figure 10: Examples of unimal agents, adapted from Gupta et al. [44]

## C.4 Comparison to Existing Benchmark

Since the current RL community has not paid much attention to embodied control so far, there are no suitable benchmarks to quantify the generalization over various tasks beyond single locomotion tasks and morphologies at the same time. In addition, the scalability to various morphologies or tasks seems to be required for the benchmark, because we should avoid "overfitting" to manually-designed tasks.

As summarized in Figure 4, MuJoCo [96] or DM Control [94], the most popular benchmarks in the continuous control domain, could not evaluate task or morphology generalization; they only have a single morphology for a single task as a pre-defined environment. Yu et al. [103] propose a robot manipulation benchmark for meta RL, but it does not care about the morphology. Furthermore, while it has quite a diverse set of tasks, the scalability of environments seems to be limited. In contrast, previous morphology-agnostic RL works [49, 62, 48, 97] have a set of different morphologies adapted from MuJoCo agents. Gupta et al. [44] also provide a much larger set of agents that are produced via joint-optimization of morphology and task rewards by the evolutionary strategy [43], with kinematics and dynamics randomization. However, those works only aim to solve single locomotion tasks, i.e. running forward as fast as possible.

# D Details of Expert Data Generation

We train the single-task PPO [89] at each environment to obtain the expert policy. For the reward function, we basically adopt dense reward (except for push task) $-d_\psi(\boldsymbol{s}^m, s_g)$ until agents satisfy a given condition, $\mathbb{1}[d_\psi(\boldsymbol{s}^m, s_g) \leq d_{\min}^{m,\psi}]$. See Appendix E for the threshold $d_{\min}^{m,\psi}$. After convergence, we collect the proficient behaviors. Unless otherwise specified, we use 12k transitions per environment. The average normalized final distance of those datasets are mostly less than 0.1.

# E Environments Division

Throughout the experiments, we test a lot of morphology-task combinations to investigate the in-distribution generalization, compositional generalization for morphology and task, and out-of-distribution generalization. In this section, we list up the combination of environments used in the experiments.

Table 4 explains the combinations for the experiments of in-distribution generalization, and compositional generalization for morphology (both zero-shot transfer and fine-tuning) in Table 1 and Figure 5. For compositional morphology evaluation, we use Morph-Train division as training dataset and Morph-Test division as evaluation environments. For the evaluation of dataset size and the number of morphology-task combinations (Appendix H), we use In-Distribution, 12 Env, and 25 Env division as train datasets and test environments.

Table 6 also explains the combinations for the experiments of in-distribution generalization with unimals [44] in Table 1 and Table 11.

Table 5 provides the combinations for the experiments of compositional generalization for task and out-of-distribution generalization (both zero-shot transfer and fine-tuning) in Table 1 and Figure 5. We use them as test environments, and for training datasets, we leverage In-Distribution division in Table 4.

In addition, we extensively evaluate the compositional generalization for task and out-of-distribution generalization with more different unseen task, such as push (see Appendix L for the details). Table 7 also shows the environment division for both zero-shot transfer and fine-tuning.

| Sub-domain | Environment | $d_{\min}^\psi$ | $d_{\max}^\psi$ | In-Distribution | Morph-Train | Morph-Test | 12 Env | 25 Env |
|---|---|---|---|---|---|---|---|---|
| ant_reach | ant_reach_2 | 0.1 | 8.75 | ✔ | ✔ | | | |
| | ant_reach_3 | 0.1 | 8.75 | ✔ | ✔ | | | |
| | ant_reach_4 | 0.1 | 8.75 | ✔ | | ✔ | ✔ | ✔ |
| | ant_reach_5 | 0.1 | 8.75 | ✔ | ✔ | | | |
| | ant_reach_6 | 0.1 | 8.75 | ✔ | ✔ | | | ✔ |
| ant_touch | ant_touch_3 | 0.5 | 3.5 | ✔ | ✔ | | | ✔ |
| | ant_touch_4 | 0.5 | 3.5 | ✔ | | ✔ | ✔ | ✔ |
| | ant_touch_5 | 0.5 | 3.5 | ✔ | ✔ | | | |
| | ant_touch_6 | 0.5 | 3.5 | ✔ | ✔ | | | ✔ |
| ant_twisters | ant_reach2_handsup_4 | {0.1, 0.1, 0.1} | {1.4, 1.4, 0.625} | ✔ | | ✔ | | |
| | ant_reach2_handsup_5 | {0.1, 0.1, 0.1} | {1.4, 1.4, 0.625} | ✔ | ✔ | | ✔ | ✔ |
| | ant_reach2_handsup_6 | {0.1, 0.1, 0.1} | {1.4, 1.4, 0.625} | ✔ | ✔ | | | ✔ |
| | ant_reach_handsup2_3 | {0.1, 0.1, 0.1} | {1.5, 0.55, 0.55} | ✔ | ✔ | | | |
| | ant_reach_handsup2_4 | {0.1, 0.1, 0.1} | {1.5, 0.55, 0.55} | ✔ | | ✔ | | ✔ |
| | ant_reach_handsup2_5 | {0.1, 0.1, 0.1} | {1.5, 0.55, 0.55} | ✔ | ✔ | | ✔ | ✔ |
| | ant_reach_handsup2_6 | {0.1, 0.1, 0.1} | {1.5, 0.55, 0.55} | ✔ | ✔ | | | |
| | ant_reach_handsup_4 | {0.1, 0.1} | {4.5, 0.55} | ✔ | | ✔ | | ✔ |
| | ant_reach_handsup_5 | {0.1, 0.1} | {4.5, 0.55} | ✔ | ✔ | | ✔ | ✔ |
| | ant_reach_handsup_6 | {0.1, 0.1} | {4.5, 0.55} | ✔ | ✔ | | | |
| claw_reach | claw_reach_2 | 0.1 | 8.75 | ✔ | ✔ | | | ✔ |
| | claw_reach_3 | 0.1 | 8.75 | ✔ | ✔ | | | |
| | claw_reach_4 | 0.1 | 8.75 | ✔ | ✔ | | ✔ | |
| | claw_reach_5 | 0.1 | 8.75 | ✔ | | ✔ | | ✔ |
| | claw_reach_6 | 0.1 | 8.75 | ✔ | ✔ | | | |
| claw_touch | claw_touch_3 | 0.5 | 3.5 | ✔ | ✔ | | | |
| | claw_touch_4 | 0.5 | 3.5 | ✔ | ✔ | | ✔ | ✔ |
| | claw_touch_5 | 0.5 | 3.5 | ✔ | | ✔ | | |
| | claw_touch_6 | 0.5 | 3.5 | ✔ | ✔ | | | ✔ |
| claw_twisters | claw_reach_handsup_3 | {0.1, 0.1} | {1.4, 0.625} | ✔ | ✔ | | | ✔ |
| | claw_reach_handsup_4 | {0.1, 0.1} | {1.4, 0.625} | ✔ | ✔ | | | |
| | claw_reach_handsup_5 | {0.1, 0.1} | {1.4, 0.625} | ✔ | | ✔ | ✔ | ✔ |
| | claw_reach_handsup_6 | {0.1, 0.1} | {1.4, 0.625} | ✔ | ✔ | | | |
| | claw_reach_hard_handsup_5 | {0.1, 0.1} | {4.5, 0.55} | ✔ | | ✔ | ✔ | ✔ |
| | claw_reach_hard_handsup_6 | {0.1, 0.1} | {4.5, 0.55} | ✔ | ✔ | | | |
| centipede_reach | centipede_reach_2 | 0.1 | 3.0 | ✔ | ✔ | | | |
| | centipede_reach_3 | 0.1 | 3.0 | ✔ | ✔ | | ✔ | ✔ |
| | centipede_reach_4 | 0.1 | 3.0 | ✔ | ✔ | | | |
| | centipede_reach_5 | 0.1 | 3.0 | ✔ | ✔ | | | |
| | centipede_reach_6 | 0.1 | 3.0 | ✔ | | ✔ | | |
| | centipede_reach_7 | 0.1 | 3.0 | ✔ | ✔ | | | ✔ |
| centipede_touch | centipede_touch_3 | 0.5 | 10.5 | ✔ | ✔ | | | |
| | centipede_touch_4 | 0.5 | 10.5 | ✔ | ✔ | | | ✔ |
| | centipede_touch_5 | 0.5 | 10.5 | ✔ | ✔ | | | ✔ |
| | centipede_touch_6 | 0.5 | 10.5 | ✔ | | ✔ | ✔ | ✔ |
| | centipede_touch_7 | 0.5 | 10.5 | ✔ | ✔ | | | |
| worm_touch | worm_touch_3 | 0.5 | 3.5 | ✔ | ✔ | | | |
| | worm_touch_4 | 0.5 | 3.5 | ✔ | ✔ | | | ✔ |
| | worm_touch_5 | 0.5 | 3.5 | ✔ | | ✔ | ✔ | ✔ |
| | worm_touch_6 | 0.5 | 3.5 | ✔ | ✔ | | | ✔ |
| | worm_touch_7 | 0.5 | 3.5 | ✔ | ✔ | | | |

Table 4: The combinations of environments used in the experiments of in-distribution generalization, and compositional generalization for morphology.

| Sub-domain | Environment | $d_{\min}^{\psi}$ | $d_{\max}^{\psi}$ | Task-Test | OOD-Test |
|---|---|---|---|---|---|
| ant_reach_hard | ant_reach_hard_3 | 0.1 | 11.0 | ✔ | |
| | ant_reach_hard_4 | 0.1 | 11.0 | ✔ | |
| | ant_reach_hard_5 | 0.1 | 11.0 | ✔ | |
| | ant_reach_hard_6 | 0.1 | 11.0 | ✔ | |
| centipede_reach_hard | centipede_reach_hard_3 | 0.1 | 5.5 | ✔ | |
| | centipede_reach_hard_4 | 0.1 | 5.5 | ✔ | |
| | centipede_reach_hard_5 | 0.1 | 5.5 | ✔ | |
| | centipede_reach_hard_6 | 0.1 | 5.5 | ✔ | |
| | centipede_reach_hard_7 | 0.1 | 5.5 | ✔ | |
| ant_reach_hard_diverse | ant_reach_hard_4_b | 0.1 | 11.0 | | ✔ |
| | ant_reach_hard_5_b | 0.1 | 11.0 | | ✔ |
| | ant_reach_hard_4_mass_0.5_1.0_3.0 | 0.1 | 11.0 | | ✔ |
| | ant_reach_hard_4_mass_0.5_1.0_1.0 | 0.1 | 11.0 | | ✔ |
| | ant_reach_hard_4_mass_1.0_3.0_3.0 | 0.1 | 11.0 | | ✔ |
| | ant_reach_hard_4_size_0.9_1.0_1.1 | 0.1 | 11.0 | | ✔ |
| | ant_reach_hard_5_mass_0.5_1.0_3.0 | 0.1 | 11.0 | | ✔ |
| | ant_reach_hard_5_mass_0.5_1.0_1.0 | 0.1 | 11.0 | | ✔ |
| | ant_reach_hard_5_mass_1.0_3.0_3.0 | 0.1 | 11.0 | | ✔ |
| | ant_reach_hard_5_size_0.9_1.0_1.1 | 0.1 | 11.0 | | ✔ |
| centipede_reach_hard_diverse | centipede_reach_hard_3_b_r_0 | 0.1 | 5.5 | | ✔ |
| | centipede_reach_hard_3_b_l_0 | 0.1 | 5.5 | | ✔ |
| | centipede_reach_hard_3_b_r_1 | 0.1 | 5.5 | | ✔ |
| | centipede_reach_hard_3_b_l_1 | 0.1 | 5.5 | | ✔ |
| | centipede_reach_hard_4_b_r_0 | 0.1 | 5.5 | | ✔ |
| | centipede_reach_hard_4_b_r_1 | 0.1 | 5.5 | | ✔ |
| | centipede_reach_hard_4_b_l_1 | 0.1 | 5.5 | | ✔ |
| | centipede_reach_hard_4_b_r_2 | 0.1 | 5.5 | | ✔ |
| | centipede_reach_hard_4_b_l_2 | 0.1 | 5.5 | | ✔ |
| | centipede_reach_hard_3_size_0.9_1.0_1.1 | 0.1 | 5.5 | | ✔ |
| | centipede_reach_hard_3_mass_0.5_1.0_3.0 | 0.1 | 5.5 | | ✔ |
| | centipede_reach_hard_3_mass_0.5_1.0_1.0 | 0.1 | 5.5 | | ✔ |
| | centipede_reach_hard_3_mass_1.0_3.0_3.0 | 0.1 | 5.5 | | ✔ |
| | centipede_reach_hard_4_size_0.9_1.0_1.1 | 0.1 | 5.5 | | ✔ |
| | centipede_reach_hard_4_mass_0.5_1.0_3.0 | 0.1 | 5.5 | | ✔ |
| | centipede_reach_hard_4_mass_0.5_1.0_1.0 | 0.1 | 5.5 | | ✔ |
| | centipede_reach_hard_4_mass_1.0_3.0_3.0 | 0.1 | 5.5 | | ✔ |

Table 5: The combinations of environments used in the experiments of compositional generalization for task and out-of-distribution generalization.

| unimal_id | $d_{\min}^{\psi}$ | $d_{\max}^{\psi}$ | reach | touch | twisters |
|---|---|---|---|---|---|
| 5506-0-13-17-12-26-41 | | | | | |
| 5506-12-12-01-11-30-00 | 0.1 | 8.75 | ✔ | | |
| 5506-15-16-01-14-17-18 | 0.35 | 3.5 | | ✔ | |
| 5506-6-5-01-14-20-42 | {0.1, 0.1} | {1.5, 0.55} | | | ✔ |
| 5506-0-2-01-15-36-43 | {0.1, 0.1} | {1.5, 0.55} | | | ✔ |
| 5506-12-12-01-15-33-01 | 0.1 | 8.75 | ✔ | | |
| 5506-15-16-02-22-21-06 | 0.1 | 8.75 | ✔ | | |
| 5506-6-8-17-09-59-06 | | | | | |
| 5506-0-5-01-12-45-36 | {0.1, 0.1} | {1.5, 0.55} | | | ✔ |
| 5506-12-14-01-15-22-01 | 0.1 | 8.75 | ✔ | | |
| 5506-2-0-01-11-27-44 | {0.1, 0.1} | {1.5, 0.55} | | | ✔ |
| 5506-7-6-17-12-20-01 | 0.35 | 3.5 | | ✔ | |
| 5506-0-7-01-15-34-13 | 0.35 | 3.5 | | ✔ | |
| 5506-12-6-17-08-36-18 | | | | | |
| 5506-2-16-01-10-58-23 | 0.1 | 8.75 | ✔ | | |
| 5506-8-11-01-15-28-53 | 0.35 | 3.5 | | ✔ | |
| 5506-1-12-17-11-10-12 | 0.1 | 8.75 | ✔ | | |
| 5506-12-6-17-12-20-06 | 0.1 | 8.75 | ✔ | | |
| 5506-2-17-17-10-16-02 | 0.1 | 8.75 | ✔ | | |
| 5506-8-12-01-13-32-46 | 0.35 | 3.5 | | ✔ | |
| 5506-1-13-17-12-03-16 | 0.1 | 8.75 | ✔ | | |
| 5506-12-8-01-14-50-41 | {0.1, 0.1} | {1.5, 0.55} | | | ✔ |
| 5506-2-9-17-11-10-42 | | | | | |
| 5506-8-16-01-13-07-43 | 0.1 | 8.75 | ✔ | | |
| 5506-1-15-17-07-32-47 | {0.1, 0.1} | {1.5, 0.55} | | | ✔ |
| 5506-13-10-17-12-25-45 | 0.1 | 8.75 | ✔ | | |
| 5506-3-10-01-14-19-06 | 0.1 | 8.75 | ✔ | | |
| 5506-8-16-02-14-47-12 | {0.1, 0.1} | {1.5, 0.55} | | | ✔ |
| 5506-1-2-02-20-28-11 | {0.1, 0.1} | {1.5, 0.55} | | | ✔ |
| 5506-13-17-01-16-09-18 | {0.1, 0.1} | {1.5, 0.55} | | | ✔ |
| 5506-3-15-01-14-36-50 | 0.35 | 3.5 | | ✔ | |
| 5506-8-17-17-09-38-29 | | | | | |
| 5506-1-5-02-19-23-33 | {0.1, 0.1} | {1.5, 0.55} | | | ✔ |
| 5506-13-3-02-21-34-38 | {0.1, 0.1} | {1.5, 0.55} | | | ✔ |
| 5506-3-15-17-12-18-03 | 0.1 | 8.75 | ✔ | | |
| 5506-8-5-02-21-39-20 | {0.1, 0.1} | {1.5, 0.55} | | | ✔ |
| 5506-10-0-01-15-43-53 | 0.35 | 3.5 | | ✔ | |
| 5506-13-4-02-21-40-07 | 0.1 | 8.75 | ✔ | | |
| 5506-4-12-01-15-10-52 | 0.35 | 3.5 | | ✔ | |
| 5506-8-6-01-15-22-56 | {0.1, 0.1} | {1.5, 0.55} | | | ✔ |
| 5506-10-12-02-12-35-19 | 0.1 | 8.75 | ✔ | | |
| 5506-13-5-02-21-35-41 | 0.1 | 8.75 | ✔ | | |
| 5506-4-14-01-14-32-47 | 0.1 | 8.75 | ✔ | | |
| 5506-9-12-01-10-32-52 | 0.1 | 8.75 | ✔ | | |
| 5506-10-13-01-15-03-41 | 0.35 | 3.5 | | ✔ | |
| 5506-14-11-01-13-58-37 | {0.1, 0.1} | {1.5, 0.55} | | | ✔ |
| 5506-4-16-17-05-46-47 | | | | | |
| 5506-9-2-01-14-19-00 | {0.1, 0.1} | {1.5, 0.55} | | | ✔ |
| 5506-10-14-17-10-38-34 | 0.35 | 3.5 | | ✔ | |
| 5506-14-12-01-12-02-42 | {0.1, 0.1} | {1.5, 0.55} | | | ✔ |
| 5506-4-3-01-09-35-18 | 0.35 | 3.5 | | ✔ | |
| 5506-9-3-01-14-23-39 | {0.1, 0.1} | {1.5, 0.55} | | | ✔ |
| 5506-10-3-01-15-22-34 | 0.35 | 3.5 | | ✔ | |
| 5506-14-15-01-15-20-33 | 0.35 | 3.5 | | ✔ | |
| 5506-5-12-01-15-05-55 | | | | | |
| 5506-9-7-01-13-40-02 | 0.1 | 8.75 | ✔ | | |
| 5506-10-3-17-12-09-26 | | | | | |
| 5506-14-2-02-15-14-46 | 0.1 | 8.75 | ✔ | | |
| 5506-5-16-02-21-15-42 | | | | | |
| 5506-9-9-01-13-15-48 | 0.35 | 3.5 | | ✔ | |
| 5506-11-2-01-14-11-40 | 0.35 | 3.5 | | ✔ | |
| 5506-14-5-01-15-59-52 | | | | | |
| 5506-5-3-02-18-52-53 | {0.1, 0.1} | {1.5, 0.55} | | | ✔ |
| 5506-11-4-17-12-33-10 | {0.1, 0.1} | {1.5, 0.55} | | | ✔ |
| 5506-15-11-01-10-04-14 | 0.35 | 3.5 | | ✔ | |
| 5506-6-11-01-14-16-09 | 0.35 | 3.5 | | ✔ | |
| 5506-11-6-17-12-43-05 | | | | | |
| 5506-15-11-01-12-54-35 | {0.1, 0.1} | {1.5, 0.55} | | | ✔ |
| 5506-6-2-01-09-16-44 | 0.35 | 3.5 | | ✔ | |
| 5506-12-11-17-05-56-16 | 0.35 | 3.5 | | ✔ | |
| 5506-15-11-17-12-12-28 | | | | | |
| 5506-6-3-01-15-20-20 | 0.35 | 3.5 | | ✔ | |

Table 6: Unimal IDs we adapted from Gupta et al. [44]. We inspect 100 morphologies and select 72 morphologies that work healthily. In the experiment of Table 1 and Table 11, we select 20 morphologies each for 3 tasks (reach, touch, twisters) as listed above.

| Sub-domain | Environment | $d_{\min}^{\psi}$ | $d_{\max}^{\psi}$ | Task-Test | OOD-Test |
|---|---|---|---|---|---|
| ant_push | ant_push_3 | 1.0 | 4.0 | ✔ | |
| | ant_push_4 | 1.0 | 4.0 | ✔ | |
| | ant_push_5 | 1.0 | 4.0 | ✔ | |
| | ant_push_6 | 1.0 | 4.0 | ✔ | |
| centipede_push | centipede_push_3 | 1.5 | 3.75 | ✔ | |
| | centipede_push_4 | 1.5 | 3.75 | ✔ | |
| | centipede_push_5 | 1.5 | 3.75 | ✔ | |
| | centipede_push_6 | 1.5 | 3.75 | ✔ | |
| | centipede_push_7 | 1.5 | 3.75 | ✔ | |
| worm_push | ant_push_3 | 1.5 | 3.25 | ✔ | |
| | worm_push_4 | 1.5 | 3.25 | ✔ | |
| | worm_push_5 | 1.5 | 3.25 | ✔ | |
| ant_push_diverse | ant_push_4_b | 1.0 | 4.0 | | ✔ |
| | ant_push_5_b | 1.0 | 4.0 | | ✔ |
| | ant_push_4_mass_0.5_1.0_3.0 | 1.0 | 4.0 | | ✔ |
| | ant_push_4_mass_0.5_1.0_1.0 | 1.0 | 4.0 | | ✔ |
| | ant_push_4_mass_1.0_3.0_3.0 | 1.0 | 4.0 | | ✔ |
| | ant_push_4_size_0.9_1.0_1.1 | 1.0 | 4.0 | | ✔ |
| | ant_push_5_mass_0.5_1.0_3.0 | 1.0 | 4.0 | | ✔ |
| | ant_push_5_mass_0.5_1.0_1.0 | 1.0 | 4.0 | | ✔ |
| | ant_push_5_mass_1.0_3.0_3.0 | 1.0 | 4.0 | | ✔ |
| | ant_push_5_size_0.9_1.0_1.1 | 1.0 | 4.0 | | ✔ |
| centipede_push_diverse | centipede_push_3_b_r_0 | 1.5 | 3.75 | | ✔ |
| | centipede_push_3_b_l_0 | 1.5 | 3.75 | | ✔ |
| | centipede_push_3_b_r_1 | 1.5 | 3.75 | | ✔ |
| | centipede_push_3_b_l_1 | 1.5 | 3.75 | | ✔ |
| | centipede_push_4_b_r_0 | 1.5 | 3.75 | | ✔ |
| | centipede_push_4_b_r_1 | 1.5 | 3.75 | | ✔ |
| | centipede_push_4_b_l_1 | 1.5 | 3.75 | | ✔ |
| | centipede_push_4_b_r_2 | 1.5 | 3.75 | | ✔ |
| | centipede_push_4_b_l_2 | 1.5 | 3.75 | | ✔ |
| | centipede_push_3_size_0.9_1.0_1.1 | 1.5 | 3.75 | | ✔ |
| | centipede_push_3_mass_0.5_1.0_3.0 | 1.5 | 3.75 | | ✔ |
| | centipede_push_3_mass_0.5_1.0_1.0 | 1.5 | 3.75 | | ✔ |
| | centipede_push_3_mass_1.0_3.0_3.0 | 1.5 | 3.75 | | ✔ |
| | centipede_push_4_size_0.9_1.0_1.1 | 1.5 | 3.75 | | ✔ |
| | centipede_push_4_mass_0.5_1.0_3.0 | 1.5 | 3.75 | | ✔ |
| | centipede_push_4_mass_0.5_1.0_1.0 | 1.5 | 3.75 | | ✔ |
| | centipede_push_4_mass_1.0_3.0_3.0 | 1.5 | 3.75 | | ✔ |

Table 7: The extra combinations of environments used in the experiments of compositional generalization for task and out-of-distribution generalization with push task.

# F   Representation Selection: Local Observations

In this section, we explore the set of node features that is most suitable for behavior distillation in multi-morphology-multi-task settings, and provide the design principle of node features. Previous morphology-agnostic studies have worked on searching network architectures that could efficiently process the encoded morphological graph, such as GNN [100], GNN for tree structure [49], and various Transformers [62, 44, 97, 48]. However, it is still unclear what kind of observations in the system could contribute to morphology-task generalization as the local features for each node of control graph. It could be valuable to identify how each feature contributes to goal-reaching or generalization and which sets of node features perform best.

There are a lot of observable candidates per module in the agent's system, such as Cartesian position (**p**), Cartesian velocity (**v**), quotation (**q**), angular velocity (**a**), joint angle (**ja**), joint range (**jr**), limb id (**id**), joint velocity (**jv**), relative position (**rp**), relative rotation (**rr**), and morphological information (**m**). Morphological information contains module's shape, mass, inertia, actuator's gear, and dof-index, etc. To shed light on the importance of the local representation per module, we execute an extensive ablation of node feature selections and prepossessing. In prior works, Huang et al. [49] and the following [62, 48] used {**p**, **v**, **q**, **a**, **ja**, **jr**, **id**} and Gupta et al. [44] used {**p**, **v**, **q**, **a**, **ja**, **jr**, **jv**, **rp**, **rr**, **m**}. Considering the intersection of those, we define {**p**, **v**, **q**, **a**, **ja**, **jr**} as `base_set` and test the combination to other observations (**jv**, **id**, **rp**, **rr**, **m**).

We prepare In-Distribution division in Table 4 and evaluate in-distribution generalization with CGv2 IO. Table 8 shows that, while some additional features (**id**, **rp**, **m**) contribute to improving the goal-reaching performance, the most effective feature seems morphological information (**m**). These results suggest `base_set` contains sufficient features for control, and raw morphological properties (**m**) serves better task specification than manually-encoded information such as limb id (**id**). The key observations are; (1) morphological information is critical for morphology-task generalization, and (2) extra observation might disrupt the goal-reaching performances, such as relative rotation between parent and child node (**rr**). Throughout the paper, we use `base_set-m` for node features of control graph representation.

| Node features | +jv | +id | +rp | +rr | +m | Average Dist. |
|---|---|---|---|---|---|---|
| `base_set` | | | | | | $0.4330 \pm 0.02$ |
| `base_set-id` | | ✔ | | | | $0.4090 \pm 0.02$ |
| `base_set-rp` | | | ✔ | | | $0.3820 \pm 0.01$ |
| `base_set-rr` | | | | ✔ | | $0.4543 \pm 0.01$ |
| `base_set-m` | | | | | ✔ | $\mathbf{0.3128 \pm 0.02}$ |
| `base_set-rp-rr` | | | ✔ | ✔ | | $0.3869 \pm 0.01$ |
| `base_set-jv-rp-rr` | ✔ | | ✔ | ✔ | | $0.4000 \pm 0.01$ |
| `base_set-jv-rp-rr-m` | ✔ | | ✔ | ✔ | ✔ | $0.3323 \pm 0.01$ |

Table 8: Combination of control-oriented node features. We compare the combination of Cartesian position (**p**), Cartesian velocity (**v**), quotation (**q**), angular velocity (**a**), joint angle (**ja**), joint range (**jr**), limb id (**id**), joint velocity (**jv**), relative position (**rp**), relative rotation (**rr**), and morphological information (**m**). `base_set` is composed of {**p**, **v**, **q**, **a**, **ja**, **jr**}. In prior works, Huang et al. [49], Kurin et al. [62], Hong et al. [48] used `base_set-id` and Gupta et al. [44] used `base_set-jv-rp-rr-m`. While some additional features (**id**, **rp**, **m**) contribute to improving the goal-reaching performance, the most effective feature seems morphological information. These results suggest `base_set` contains sufficient features for control. and raw morphological properties (**m**) serves better task specification than manually-encoded information such as limb id (**id**).

# G    Architecture Selection: Tokenized Control Graph

In the recent literature, offline RL is considered as supervised sequential modeling problem [12], and some works [53, 87] tokenize the continuous observations and actions, like an analogy of vision transformer [26]; treating input modality like a language.

As a part of offline architecture selection, we examine the effectiveness of tokenization. We mainly follow the protocol in Reed et al. [87]; we first apply mu-law encoding [77] to control graph representation:

$$\text{mu\_law}(x) := \text{sgn}(x)\frac{\log(|x|\mu + 1)}{\log(M\mu + 1)},$$

with $\mu = 100$ and $M = 256$. This pre-processing could normalize the input to the range of $[0, 1]$. Then, we discretize pre-processed observations and actions with 1024 bins. To examine the broader range of design choice, we prepare the following 6 variants of tokenized control graph: whether layer normalization is added after embedding function (LN) [7, 78, 34, 12], or outputting discretized action (D), smoothed action by taking the average of bins (DA), and continuous values directly (C).

As shown in Table 9, predicting continuous value reveals the better performance than discretized action maybe due to some approximation errors. However, the performance of Token-CGv2 (C) or Token-CGv2 (C, LN) is still lower than CGv2 itself. This might be because tokenization looses some morphological invariance among nodes. In Table 1, we adopt Token-CGv2 (C) for comparison.

| Sub-domain | Token-CGv2 (D, LN) | Token-CGv2 (DA, LN) | Token-CGv2 (C, LN) | Token-CGv2 (D) | Token-CGv2 (DA) | Token-CGv2 (C) | Transformer (CGv2) |
|---|---|---|---|---|---|---|---|
| ant_reach | $0.9256 \pm 0.01$ | $0.9252 \pm 0.01$ | $0.4394 \pm 0.01$ | $0.9210 \pm 0.02$ | $0.9215 \pm 0.00$ | $0.3846 \pm 0.03$ | $0.3206 \pm 0.06$ |
| ant_touch | $1.0373 \pm 0.01$ | $1.0631 \pm 0.01$ | $0.4465 \pm 0.01$ | $1.0528 \pm 0.00$ | $1.0630 \pm 0.01$ | $0.3458 \pm 0.03$ | $0.2668 \pm 0.08$ |
| ant_twisters | $0.5581 \pm 0.00$ | $0.5655 \pm 0.00$ | $0.2090 \pm 0.00$ | $0.5587 \pm 0.01$ | $0.5655 \pm 0.00$ | $0.2487 \pm 0.01$ | $0.1039 \pm 0.05$ |
| claw_reach | $0.9568 \pm 0.00$ | $0.9584 \pm 0.01$ | $0.3685 \pm 0.01$ | $0.9674 \pm 0.00$ | $0.9583 \pm 0.01$ | $0.2862 \pm 0.07$ | $0.3581 \pm 0.04$ |
| claw_touch | $1.0300 \pm 0.01$ | $1.0584 \pm 0.01$ | $0.3238 \pm 0.07$ | $1.0392 \pm 0.03$ | $1.0585 \pm 0.01$ | $0.3229 \pm 0.07$ | $0.2573 \pm 0.08$ |
| claw_twisters | $0.6228 \pm 0.01$ | $0.6205 \pm 0.00$ | $0.4035 \pm 0.02$ | $0.6241 \pm 0.00$ | $0.6202 \pm 0.00$ | $0.3810 \pm 0.03$ | $0.3442 \pm 0.04$ |
| centipede_reach | $0.5692 \pm 0.12$ | $0.5784 \pm 0.13$ | $0.1166 \pm 0.00$ | $0.6373 \pm 0.09$ | $0.5780 \pm 0.13$ | $0.1132 \pm 0.03$ | $0.1057 \pm 0.04$ |
| centipede_touch | $0.9818 \pm 0.01$ | $1.0088 \pm 0.00$ | $0.1696 \pm 0.00$ | $0.9910 \pm 0.01$ | $1.0087 \pm 0.00$ | $0.1823 \pm 0.03$ | $0.3869 \pm 0.04$ |
| worm_touch | $1.0596 \pm 0.01$ | $1.0731 \pm 0.01$ | $0.9357 \pm 0.14$ | $1.0639 \pm 0.01$ | $1.0706 \pm 0.01$ | $0.9235 \pm 0.07$ | $0.8952 \pm 0.04$ |
| **Average Dist.** | $0.8124 \pm 0.01$ | $0.8232 \pm 0.02$ | $0.3572 \pm 0.02$ | $0.8248 \pm 0.01$ | $0.8225 \pm 0.02$ | $0.3402 \pm 0.01$ | $0.3128 \pm 0.02$ |

Table 9: The average normalized final distance in in-distribution evaluation. We extensively evaluate the tokenized control graph variants, similar to Reed et al. [87].

# H   Does Transformer with Control Graph scale up with dataset/morphology-task/model size?

The important aspect of the success in large language models is the scalability to the size of training data, the number of tasks for joint-training, and the number of parameters [86, 10]. One natural question is whether the similar trend holds even in RL.

Figure 11 suggests that the performance can get better if we increase the number of datasets and the number of parameters. The performance of Transformer with 0.4M parameters is equivalent to that of MLP with 3.1M. In contrast, when we increase the number of environments, the performance degrades while Transformer withCGv1 and CGv2 surpasses the degree of degradation, which seems inevitable trends in multi-task RL [104, 63] and an important future direction towards generalist controllers.

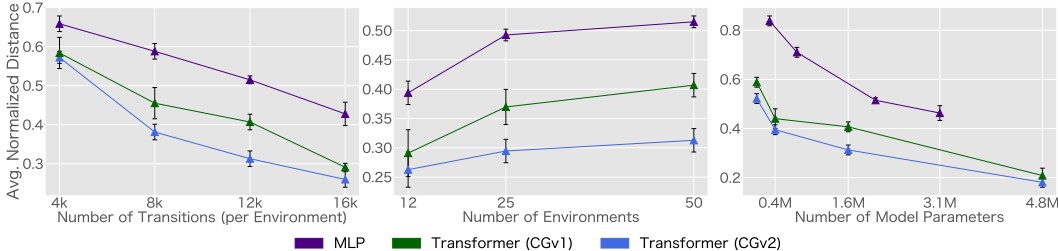

Figure 11: The average normalized final distance with different size of datasets (left), morphology-task combinations (middle), and model size (right). The smaller value means the better multi-task performance (see Appendix E for environment division). These results suggest that the performance can get better when we increase the number of datasets, and control graph can surpasses the degradation of the performance when we increase the number of environments. Transformer is parameter-efficient than MLP, and improves the performance as many parameters.

# I   Percentage of Improvement

For clarification, we compute the percentage of improvement between two average normalized final distances (defined in Equation 3) $\bar{d}_1$ and $\bar{d}_2$ as follows ($\bar{d}_1 < \bar{d}_2$):

$$100 * \frac{\bar{d}_2 - \bar{d}_1}{\bar{d}_2}.$$

# J Additional Results

In this section, we provide the detailed performance of in-distribution generalization (Table 10 and Table 11), compositional morphology and task generalization (Table 12 and Table 13), and out-of-distribution generalization (Table 14). For fine-tuning experiments, we summarized the detailed scores of Figure 5 in Table 15.

| Sub-domain | Random | MLP | GNN (CGv1) | Transformer (CGv1) | Transformer (CGv2) |
|---|---|---|---|---|---|
| ant_reach | $0.9637 \pm 0.02$ | $0.6426 \pm 0.03$ | $0.6240 \pm 0.03$ | $0.3657 \pm 0.04$ | $0.3206 \pm 0.06$ |
| ant_touch | $1.0817 \pm 0.02$ | $0.3689 \pm 0.02$ | $0.4434 \pm 0.03$ | $0.1140 \pm 0.03$ | $0.2668 \pm 0.08$ |
| ant_twisters | $0.9113 \pm 0.84$ | $0.3708 \pm 0.01$ | $0.2513 \pm 0.01$ | $0.2517 \pm 0.02$ | $0.1039 \pm 0.05$ |
| claw_reach | $1.0760 \pm 0.20$ | $0.6617 \pm 0.00$ | $0.6214 \pm 0.02$ | $0.7158 \pm 0.03$ | $0.3581 \pm 0.04$ |
| claw_touch | $1.5240 \pm 0.88$ | $0.6824 \pm 0.06$ | $0.3135 \pm 0.06$ | $0.6121 \pm 0.04$ | $0.2573 \pm 0.08$ |
| claw_twisters | $1.3786 \pm 1.42$ | $0.4907 \pm 0.03$ | $0.6063 \pm 0.07$ | $0.4614 \pm 0.07$ | $0.3442 \pm 0.04$ |
| centipede_reach | $0.5843 \pm 0.35$ | $0.0803 \pm 0.02$ | $0.1088 \pm 0.02$ | $0.0981 \pm 0.03$ | $0.1057 \pm 0.04$ |
| centipede_touch | $1.0077 \pm 0.01$ | $0.4743 \pm 0.03$ | $0.5089 \pm 0.01$ | $0.4609 \pm 0.07$ | $0.3869 \pm 0.04$ |
| worm_touch | $2.7087 \pm 1.73$ | $1.1034 \pm 0.06$ | $1.0559 \pm 0.04$ | $0.7708 \pm 0.03$ | $0.8952 \pm 0.05$ |
| **Average Dist.** | $1.2019 \pm 0.41$ | $0.5150 \pm 0.01$ | $0.4776 \pm 0.01$ | $0.4069 \pm 0.02$ | $\mathbf{0.3128 \pm 0.02}$ |

Table 10: The average normalized final distance for in-distribution evaluation on MxT-Bench (as shown in Table 1).

| Sub-domain | Random | MLP | Transformer (CGv1) | Transformer (CGv2) |
|---|---|---|---|---|
| unimal_reach | $0.9662 \pm 0.01$ | $0.7448 \pm 0.02$ | $0.5692 \pm 0.01$ | $0.4532 \pm 0.01$ |
| unimal_touch | $1.1302 \pm 0.10$ | $0.7634 \pm 0.03$ | $0.5290 \pm 0.04$ | $0.4461 \pm 0.05$ |
| unimal_twisters | $0.6305 \pm 0.02$ | $0.5027 \pm 0.02$ | $0.3534 \pm 0.03$ | $0.3540 \pm 0.02$ |
| **Average Dist.** | $0.9090 \pm 0.03$ | $0.6703 \pm 0.01$ | $0.4839 \pm 0.02$ | $\mathbf{0.4178 \pm 0.01}$ |

Table 11: The average normalized final distance for in-distribution evaluation on MxT-Bench with challenging morphologies from Gupta et al. [44] (as shown in Table 1).

| Sub-domain | Random | MLP | Transformer (CGv1) | Transformer (CGv2) |
|---|---|---|---|---|
| ant_reach | $0.9697 \pm 0.03$ | $0.8541 \pm 0.03$ | $0.5511 \pm 0.10$ | $0.4170 \pm 0.04$ |
| ant_touch | $1.0821 \pm 0.00$ | $0.7742 \pm 0.07$ | $0.4464 \pm 0.07$ | $0.3752 \pm 0.12$ |
| ant_twisters | $0.9215 \pm 0.95$ | $0.5356 \pm 0.02$ | $0.2569 \pm 0.04$ | $0.2608 \pm 0.02$ |
| claw_reach | $0.9915 \pm 0.04$ | $0.9370 \pm 0.01$ | $0.7332 \pm 0.03$ | $0.4399 \pm 0.06$ |
| claw_touch | $1.0695 \pm 0.02$ | $1.0283 \pm 0.02$ | $0.7074 \pm 0.10$ | $0.1375 \pm 0.01$ |
| claw_twisters | $0.7001 \pm 0.17$ | $0.5456 \pm 0.02$ | $0.5486 \pm 0.02$ | $0.5589 \pm 0.05$ |
| centipede_reach | $0.7929 \pm 0.04$ | $0.4626 \pm 0.15$ | $0.2640 \pm 0.07$ | $0.2506 \pm 0.14$ |
| centipede_touch | $1.0121 \pm 0.01$ | $0.7632 \pm 0.02$ | $0.4971 \pm 0.06$ | $0.3474 \pm 0.09$ |
| worm_touch | $2.7376 \pm 2.23$ | $1.1417 \pm 0.10$ | $0.8608 \pm 0.07$ | $1.0110 \pm 0.03$ |
| **Average Dist.** | $1.1419 \pm 0.41$ | $0.7216 \pm 0.01$ | $0.4940 \pm 0.01$ | $\mathbf{0.4066 \pm 0.01}$ |

Table 12: The average normalized final distance for compositional morphology evaluation on MxT-Bench (as shown in Table 1).

| Sub-domain | Random | MLP | Transformer (CGv1) | Transformer (CGv2) |
|---|---|---|---|---|
| ant_reach_hard | $0.9542 \pm 0.01$ | $0.8299 \pm 0.03$ | $0.6176 \pm 0.07$ | $0.4522 \pm 0.06$ |
| centipede_reach_hard | $0.8443 \pm 0.02$ | $0.5689 \pm 0.03$ | $0.4770 \pm 0.05$ | $0.4412 \pm 0.05$ |
| **Average Dist.** | $0.8932 \pm 0.01$ | $0.6849 \pm 0.01$ | $0.5395 \pm 0.04$ | $\mathbf{0.4461 \pm 0.05}$ |

Table 13: The average normalized final distance for compositional task evaluation on MxT-Bench (as shown in Table 1).

| Sub-domain | Random | MLP | Transformer (CGv1) | Transformer (CGv2) |
|---|---|---|---|---|
| ant_reach_hard_diverse | $0.9520 \pm 0.01$ | $0.8288 \pm 0.02$ | $0.6798 \pm 0.04$ | $0.5365 \pm 0.05$ |
| centipede_reach_hard_diverse | $0.8660 \pm 0.02$ | $0.7546 \pm 0.02$ | $0.6130 \pm 0.03$ | $0.5208 \pm 0.04$ |
| **Average Dist.** | $0.8979 \pm 0.01$ | $0.7821 \pm 0.02$ | $0.6144 \pm 0.04$ | $\mathbf{0.5266 \pm 0.04}$ |

Table 14: The average normalized final distance for out-of-distribution evaluation on MxT-Bench (as shown in Table 1). The agents are diversified with missing, mass, size randomization.

| Method (data size) | Compositional (Morphology) | Compositional (Task) | Out-of-Distribution |
|---|---|---|---|
| MLP (4K, randinit) | $0.8162 \pm 0.02$ | $0.7046 \pm 0.01$ | $0.6674 \pm 0.03$ |
| MLP (4K, fine-tuning) | $0.6715 \pm 0.02$ | $0.5582 \pm 0.03$ | $0.5694 \pm 0.04$ |
| Transformer (CGv1) (4K, randinit) | $0.6021 \pm 0.03$ | $0.5017 \pm 0.04$ | $0.5222 \pm 0.06$ |
| Transformer (CGv1) (4K, fine-tuning) | $0.3499 \pm 0.04$ | $0.3378 \pm 0.08$ | $0.3401 \pm 0.08$ |
| Transformer (CGv2) (4K, randinit) | $0.5504 \pm 0.03$ | $0.4894 \pm 0.03$ | $0.4980 \pm 0.05$ |
| Transformer (CGv2) (4K, fine-tuning) | $0.2863 \pm 0.02$ | $0.3180 \pm 0.07$ | $0.3115 \pm 0.07$ |
| MLP (8K, randinit) | $0.7044 \pm 0.02$ | $0.4818 \pm 0.02$ | $0.5247 \pm 0.04$ |
| MLP (8K, fine-tuning) | $0.5579 \pm 0.06$ | $0.3709 \pm 0.01$ | $0.4238 \pm 0.05$ |
| Transformer (CGv1) (8K, randinit) | $0.4001 \pm 0.04$ | $0.2791 \pm 0.06$ | $0.2762 \pm 0.04$ |
| Transformer (CGv1) (8K, fine-tuning) | $0.2600 \pm 0.04$ | $0.1926 \pm 0.03$ | $0.2089 \pm 0.03$ |
| Transformer (CGv2) (8K, randinit) | $0.3758 \pm 0.03$ | $0.1868 \pm 0.05$ | $0.1868 \pm 0.05$ |
| Transformer (CGv2) (8K, fine-tuning) | $0.1871 \pm 0.01$ | $0.1356 \pm 0.03$ | $0.1498 \pm 0.03$ |
| MLP (12K, randinit) | $0.5530 \pm 0.01$ | $0.4256 \pm 0.03$ | $0.4399 \pm 0.02$ |
| MLP (12K, fine-tuning) | $0.4312 \pm 0.03$ | $0.2982 \pm 0.02$ | $0.3358 \pm 0.02$ |
| Transformer (CGv1) (12K, randinit) | $0.2914 \pm 0.03$ | $0.1555 \pm 0.04$ | $0.1759 \pm 0.03$ |
| Transformer (CGv1) (12K, fine-tuning) | $0.2042 \pm 0.01$ | $0.0888 \pm 0.01$ | $0.0894 \pm 0.01$ |
| Transformer (CGv2) (12K, randinit) | $0.2655 \pm 0.03$ | $0.1149 \pm 0.02$ | $0.1149 \pm 0.02$ |
| Transformer (CGv2) (12K, fine-tuning) | $0.1301 \pm 0.02$ | $0.0562 \pm 0.02$ | $0.0513 \pm 0.01$ |

Table 15: The average normalized final distance among test environments in fine-tuning settings (compositional morphology/task or out-of-distribution evaluation).

# K   Additional Attention Analysis

In this section, we provide the full results of attention analysis of Transformer (with CGv1 and CGv2) (Figure 12 and Figure 13).

The experimental results reveal that despite slight differences, CGv2 generalizes various morphologies and tasks better than CGv1. To find out the difference between those, we qualitatively analyze the attention weights in Transformer. Figure 12 shows that CGv2 consistently focuses on goal nodes over time, and activates important nodes to solve the task; for instance, in centipede_touch (top), CGv2 pays attention to corresponding nodes (torso0 an goal0) at the beginning of the episode, and gradually sees other relevant nodes (torso1 and torso2) to hold the movable ball. Furthermore, in ant_twisters (bottom), CGv2 firstly tries to raise the agent's legs to satisfy goal1 and goal2, and then focus on reaching a leg (goal0). Temporally-consistent attention to goal nodes and dynamics attention to relevant nodes can contribute to generalization over goal-directed tasks and morphologies.

Figure 13 implies that CGv1 does not show such consistent activation to the goal-conditioned node; for instance, in centipede_touch_3 (above), the goal information is treated as an extra node feature of torso0, but there are no nodes that consistently activated with torso0. Moreover, in ant_reach_handsup2_3 (bottom), CGv1 does not keep focusing on the agent's limbs during the episode. Rather, CGv1 tends to demonstrate some periodic patterns during the rollout as implied in prior works [49, 62].

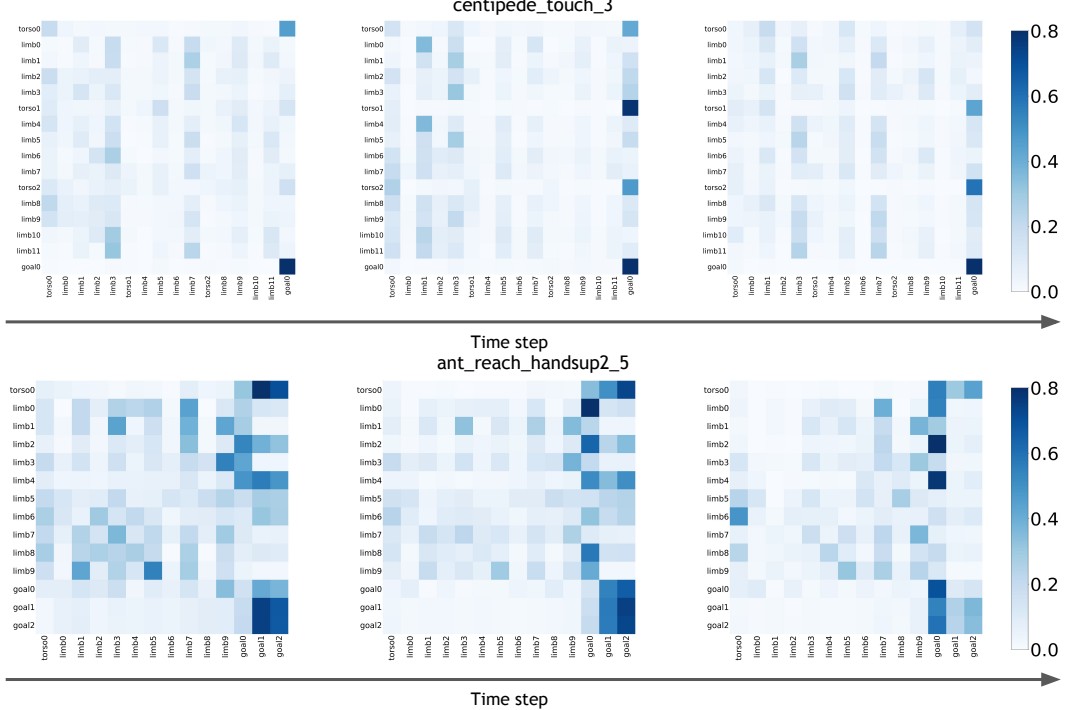

Figure 12: Attention analysis of CGv2 in centipede_touch_3 (top) and ant_reach_handsup2_5 (bottom; from twisters). From left to right, we visualize the attention weights of CGv2 during the rollout. In contrast to CGv1, CGv2 consistently focuses on goal nodes over time, and activates important nodes to solve the task.

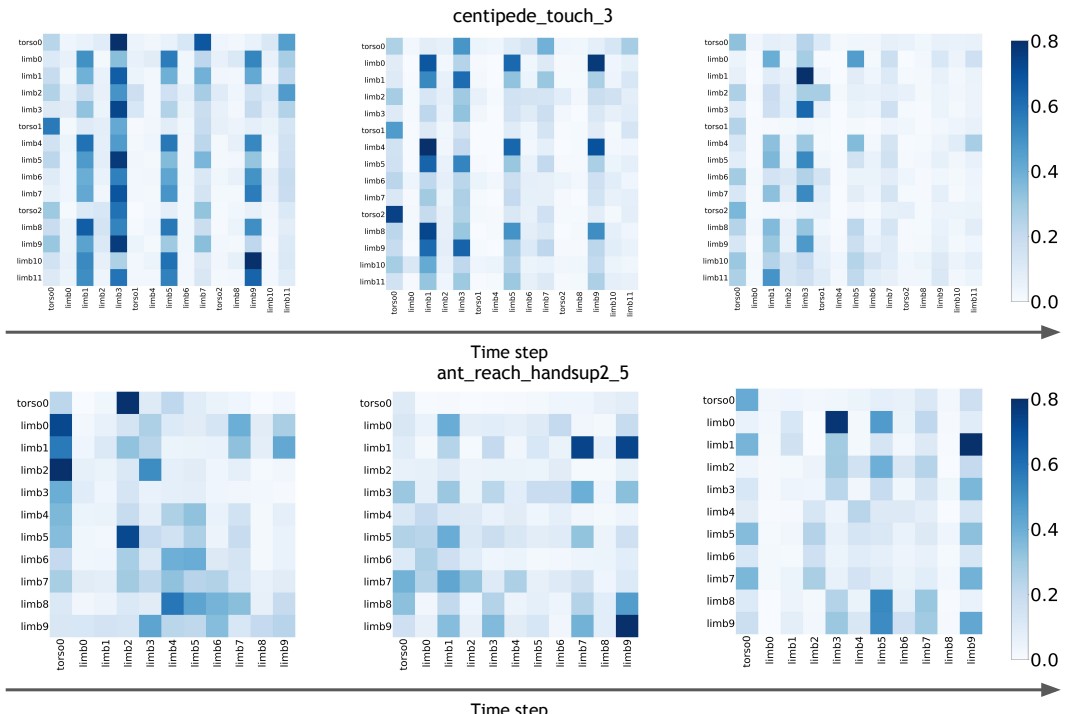

Figure 13: Attention analysis of CGv1 in centipede_touch_3 and ant_reach_handsup2_5. This tends to demonstrate some periodic patterns during the rollout as implied in prior works [49, 62].

# L Additional Results of Task Generalization

Although in Table 1 and Figure 5, we examine the compositional task generalization and out-of-distribution generalization with reach_hard tasks, where the goal distribution is farther than original reach tasks. While they seem more difficult unseen tasks [35], they also seem to have some sort of task similarities between training dataset environments and those evaluation environments.

Another question might be how control graph performs in more different unseen tasks. To evaluate compositional task generalization and out-of-distribution on the environments that have less similarity to the training datasets, we prepare push task, where the agents try to move the box objects to the given goal position. See Table 7 for the environment division. For training datasets, we leverage In-Distribution division in Table 4. Because this task requires the sufficient interaction with the object, the nature of tasks seem quite different from training dataset environments (reach, touch, and twisters).

Table 16 shows the results of compositional task evaluation and Table 17 shows those of out-of-distribution evaluation. In contrast to Table 1, the zero-shot performance seems limited. Transferring pre-trained control primitives to significantly different tasks still remains as important future work.

However, as shown in Figure 14, control graph works as better prior knowledge for downstream multi-task imitation learning, even with the environments that have less similarity to the pre-training datasets. As prior work suggested [73], these results suggests that, in RL, jointly-learned multi-task model has a strong inductive bias even for unseen and significantly different environments.

| Sub-domain | Random | MLP | Transformer (CGv1) | Transformer (CGv2) |
|---|---|---|---|---|
| ant_push | $1.0808 \pm 0.02$ | $0.9995 \pm 0.00$ | $0.9212 \pm 0.14$ | $0.8836 \pm 0.09$ |
| centipede_push | $1.0816 \pm 0.01$ | $0.9942 \pm 0.00$ | $0.9557 \pm 0.02$ | $0.9377 \pm 0.01$ |
| worm_push | $1.1026 \pm 0.04$ | $0.6300 \pm 0.01$ | $0.5579 \pm 0.10$ | $0.4091 \pm 0.15$ |
| **Average Dist.** | $1.0866 \pm 0.02$ | $0.9051 \pm 0.00$ | $0.8800 \pm 0.02$ | $0.8263 \pm 0.04$ |

Table 16: The average normalized final distance for compositional task evaluation on MxT-Bench with unseen push task. See Table 7 for the environment division.

| Sub-domain | Random | MLP | Transformer (CGv1) | Transformer (CGv2) |
|---|---|---|---|---|
| ant_push_diverse | $1.0814 \pm 0.02$ | $1.0018 \pm 0.01$ | $0.9906 \pm 0.01$ | $0.8800 \pm 0.04$ |
| centipede_push_diverse | $1.0848 \pm 0.02$ | $1.0031 \pm 0.01$ | $0.9997 \pm 0.00$ | $0.9355 \pm 0.04$ |
| **Average Dist.** | $1.0835 \pm 0.02$ | $1.0021 \pm 0.02$ | $0.9965 \pm 0.01$ | $\mathbf{0.9155 \pm 0.03}$ |

Table 17: The average normalized final distance for out-of-distribution evaluation on MxT-Bench with unseen push task. See Table 7 for the environment division.

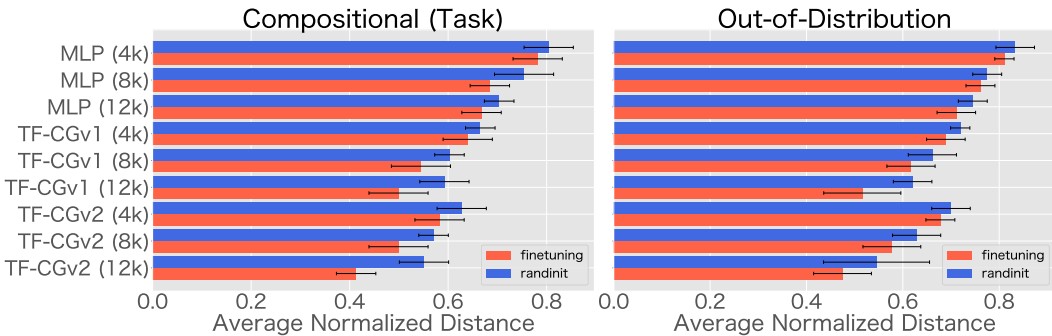

Figure 14: Comparison of multi-task goal-reaching performances on fine-tuning settings in the unseen push tasks. The results imply that control graph works as better prior knowledge for downstream multi-task imitation learning, even with the environments that have less similarity to the pre-training datasets.

| Method (data size) | Compositional (Task) | Out-of-Distribution |
|---|---|---|
| MLP (4K, randinit) | $0.8042 \pm 0.05$ | $0.8332 \pm 0.04$ |
| MLP (4K, fine-tuning) | $0.7818 \pm 0.05$ | $0.8108 \pm 0.02$ |
| Transformer (CGv1) (4K, randinit) | $0.6653 \pm 0.03$ | $0.7191 \pm 0.02$ |
| Transformer (CGv1) (4K, fine-tuning) | $0.6398 \pm 0.05$ | $0.6893 \pm 0.04$ |
| Transformer (CGv2) (4K, randinit) | $0.6276 \pm 0.05$ | $0.6999 \pm 0.04$ |
| Transformer (CGv2) (4K, fine-tuning) | $0.5826 \pm 0.05$ | $0.6779 \pm 0.03$ |
| MLP (8K, randinit) | $0.7542 \pm 0.06$ | $0.7751 \pm 0.03$ |
| MLP (8K, fine-tuning) | $0.6847 \pm 0.04$ | $0.7611 \pm 0.03$ |
| Transformer (CGv1) (8K, randinit) | $0.6027 \pm 0.03$ | $0.6613 \pm 0.05$ |
| Transformer (CGv1) (8K, fine-tuning) | $0.5448 \pm 0.06$ | $0.6169 \pm 0.05$ |
| Transformer (CGv2) (8K, randinit) | $0.5704 \pm 0.03$ | $0.6286 \pm 0.05$ |
| Transformer (CGv2) (8K, fine-tuning) | $0.4993 \pm 0.06$ | $0.5772 \pm 0.06$ |
| MLP (12K, randinit) | $0.7037 \pm 0.03$ | $0.7452 \pm 0.03$ |
| MLP (12K, fine-tuning) | $0.6678 \pm 0.04$ | $0.7110 \pm 0.06$ |
| Transformer (CGv1) (12K, randinit) | $0.5925 \pm 0.04$ | $0.6203 \pm 0.04$ |
| Transformer (CGv1) (12K, fine-tuning) | $0.4993 \pm 0.06$ | $0.5158 \pm 0.08$ |
| Transformer (CGv2) (12K, randinit) | $0.5510 \pm 0.05$ | $0.5454 \pm 0.11$ |
| Transformer (CGv2) (12K, fine-tuning) | $0.4132 \pm 0.04$ | $0.4748 \pm 0.06$ |

Table 18: The average normalized final distance among test environments in fine-tuning settings (compositional morphology/task or out-of-distribution evaluation) with unseen push task.

