# OpenReview forum: "Control Graph as Unified IO for Morphology-Task Generalization"
_NeurIPS.cc/2022/Workshop/Offline_RL — Offline RL Workshop NeurIPS 2022_

### Official Review · Reviewer_Gumi · 2022-10-19
**Many good contributions for multi-morphology RL**

**Rating:** 8
**Confidence:** 4

**Review:**

Before the review, it is worth mentioning that the paper does not follow the [submission guidelines](https://offline-rl-neurips.github.io/2022/submit.html) presented on the workshop site. More specifically, the guidelines state that “We shall accept submissions of up to 4-5 pages of content (with no limit on references and appendix)”, but the paper presents 9 pages. I will leave the decision of enforcing this to the workshop organizers and solely review the content of the paper.



Summary: This paper studies the problem of multi-task, multi-morphology RL via behavior distillation. It introduces a novel benchmark (MxT-Bench) that comprises scalable procedural generation of tasks and morphologies. It also introduces control graph, an interface that represents the agent alongside the goals, observations, and actions in the same graph space. Finally, the work studies the generalization capabilities of such representation in the MxT-Bench through behavior distillation, showing gains in multi-task performance using the transformer architecture.

Strengths:
- This work has several contributions: a benchmark for multi-task, multi-morphology RL; a new, morphology-agnostic representation for agents; and a systematic evaluation of such representation in the benchmark via behavior distillation. Therefore, the content is very rich and really advances this research topic;

- The work is well-placed in previous literature and does a decent job comparing MxT-Bench and previous benchmarks.

- The control graph also seems to be a strong contribution towards a unified representation of different agent morphologies.

Major Concerns:
- No major concerns about this work.


Minor concerns/suggestions:
- It would be interesting if the MxT-Bench could evolve for more diversified morphologies, such as cheetahs, hoppers, or even humanoids. These are complex morphologies and will validate if the control graph representation is scalable. Additionally, they are present in previous benchmarks for meta-RL and standard RL. In the same line, the benchmark could bring “velocity” tasks (locomotion with a target velocity), besides reach/touch/twister.

- The paper clearly states the gains of using a unified representation like the control graph over previous approaches. Nevertheless, one question remains: is this representation good for RL algorithms? I would suggest having some ablation that compares the learning efficiency considering the control graph and the standard scenario.

- The paper claims that the transformer architecture is better than other baselines. It would be interesting if the number of parameters are also presented in the paper. It would help refute the hypothesis that the capacity of the network is the main reason for this, rather than the associated inductive biases.

Besides these suggestions, the work has a clear impact that is aligned with workshop goals.

---

### Official Review · Reviewer_LY7g · 2022-10-20

**Rating:** 8
**Confidence:** 4

**Review:**

This paper investigates a multi-task reinforcement learning setting that involves generalizing to new morphologies, tasks, and goal spaces.  The paper proposes the first benchmark for evaluating this form of multi-task RL, noting that prior works have studied each category of multi-task RL separately, and not all at once. In addition to proposing a new benchmark, which is by itself an important contribution, this paper proposes a graph-based architecture that is called "Control Graph" in the paper for extending the morphological graph used in several prior works with additional nodes that specify the task goal. Importantly, the number of goal nodes may vary, which allows their architecture to generalize to various types of goal specification, including a single goal for some tasks, and several per-limb goals for others. Overall, the paper is quite strong experimentally, and performs several necessary evaluations, including zero-shot generalization, few-shot fine-tuning, initial results on scaling laws, and several ablations. I recommend acceptance, and have several questions and suggestions:

Questions:

(1) In Figure 5, where fine-tuning CGv2 is compared to a random initialization, on what data is fine-tuning / training performed for both? Is fine-tuning performed using RL, and if so, what are the details of the optimizer being used? Is the randomly initialized baseline non-modular?

(2) In Table 1, does MLP correspond to a non-modular policy specialized to each task? Why are results from the GNN variant of CGv1 omitted from the table on all tasks other than In-Distribution?

Suggestions:

(1) Alongside Figure 6, it could help to visualize what the agent looks like at each stage of the attention pattern to better understand what gait is being learned. Figure 6 in https://www.cs.cmu.edu/~dpathak/papers/modular-rl.pdf shows a great way to do this.

(2) In Appendix G "ARCHITECTURE SELECTION: TOKENIZED CONTROL GRAPH" it would be helpful to also discuss the connection between the tokenization method used in this work, and the tokenization method used in Trabucco et al., 2022.